# 3D-MolT5: Leveraging Discrete Structural Information for Molecule-Text Modeling

**Qizhi Pei**[1,2]  **Rui Yan**[1,3†]  **Kaiyuan Gao**[4]  **Jinhua Zhu**[5]  **Lijun Wu**[6†]
[1]Gaoling School of Artificial Intelligence, Renmin University of China
[2]Engineering Research Center of Next-Generation Intelligent Search and Recommendation, Ministry of Education
[3]School of Computer Science, Wuhan University    [4]Huazhong University of Science and Technology
[5]University of Science and Technology of China    [6]Shanghai AI Laboratory
{qizhipei,ruiyan}@ruc.edu.cn   im_kai@hust.edu.cn
teslazhu@mail.ustc.edu.cn   apeterswu@gmail.com

## Abstract

The integration of molecular and natural language representations has emerged as a focal point in molecular science, with recent advancements in Language Models (LMs) demonstrating significant potential for comprehensive modeling of both domains. However, existing approaches face notable limitations, particularly in their neglect of three-dimensional (3D) information, which is crucial for understanding molecular structures and functions. While some efforts have been made to incorporate 3D molecular information into LMs using external structure encoding modules, significant difficulties remain, such as insufficient interaction across modalities in pre-training and challenges in modality alignment. To address the limitations, we propose **3D-MolT5**, a unified framework designed to model molecule in both sequence and 3D structure spaces. The key innovation of our approach lies in mapping fine-grained 3D substructure representations into a specialized 3D token vocabulary. This methodology facilitates the seamless integration of sequence and structure representations in a tokenized format, enabling 3D-MolT5 to encode molecular sequences, molecular structures, and text sequences within a unified architecture. Leveraging this tokenized input strategy, we build a foundation model that unifies the sequence and structure data formats. We then conduct joint pre-training with multi-task objectives to enhance the model's comprehension of these diverse modalities within a shared representation space. Thus, our approach significantly improves cross-modal interaction and alignment, addressing key challenges in previous work. Further instruction tuning demonstrated that our 3D-MolT5 has strong generalization ability and surpasses existing methods with superior performance in multiple downstream tasks, such as nearly 70% improvement on the molecular property prediction task compared to state-of-the-art methods. Our code is available at `https://github.com/QizhiPei/3D-MolT5`.

## 1 Introduction

Molecule plays a pivotal role in various scientific and industrial applications, spanning from pharmaceuticals to materials science (Drews, 2000; Dara et al., 2022; AI4Science & Quantum, 2023). In recent years, the development of Language Models (LMs) (Achiam et al., 2023; Touvron et al., 2023; Dubey et al., 2024) has garnered significant attention towards the joint modeling of molecule and language (Edwards et al., 2022; Zeng et al., 2022; Li et al., 2023c). LMs trained on textual descriptions of molecules can acquire comprehensive knowledge that enhances molecular understanding, thereby improving generalization to various molecule-related tasks, such as molecule-text retrieval (Zeng et al., 2022; Su et al., 2022) and molecule captioning (Edwards et al., 2022; Liu et al., 2023b; Pei et al., 2023). Language, inherently sequential, has inspired researchers to explore autoregressive pre-training of LMs for jointly modeling molecular sequences (e.g., SMILES (Weininger, 1988), SELFIES (Krenn et al., 2020; 2022)) and text sequences (Edwards et al., 2022; Zeng et al., 2022;

---

[†]Corresponding authors: Rui Yan (`ruiyan@ruc.edu.cn`) and Lijun Wu (`apeterswu@gmail.com`).

Pei et al., 2023). To incorporate 2D graph information, two primary approaches have emerged: contrastive pre-training between 2D molecular graphs and text (Edwards et al., 2021; Su et al., 2022; Seidl et al., 2023; Luo et al., 2023; Liu et al., 2023a), and alignment of 2D molecular graph encoders with LMs (Liu et al., 2023c; Cao et al., 2023) through multi-stage pre-training inspired by BLIP2 (Li et al., 2023b).

However, most existing works have overlooked the molecular 3D structure, which contains crucial stereochemical information for function-related tasks (Ruddigkeit et al., 2012; Hu et al., 2021; Li et al., 2023c; Wang et al., 2005). Few attempts (Tang et al., 2023; Li et al., 2023c; Xiao et al., 2024; Zhao et al., 2024) try to eliminate this limitation by integrating external molecular structure encoders to incorporate the 3D molecular inputs with language. Through alignment training between the external molecular structure encoders and LMs, these approaches achieve preliminary success, but notable shortcomings exist in these methods: (1) **Insufficient interaction across modalities in pre-training**: The molecular structure encoder is pre-trained separately from the text, resulting in inadequate interaction between different modalities during pre-training. (2) **Challenges in modality alignment**: The different representation spaces of pre-trained molecular encoders and LMs require alignment training. However, modality alignment is always challenging (Baltrušaitis et al., 2018) for different reasons (e.g., limited molecule-text paired data, discrete text tokens, and continuous 3D structure representation). Besides, how to assess the quality of alignment well remains unclear. (3) **Dependency on external encoder**: Though it is efficient to incorporate the pre-trained external structure encoder, the performance of the encoder can not be directly controlled within the framework, which also raises difficulty in performance alignment.

To address these limitations, inspired by the joint multi-modal modeling in Vision-Language (Team, 2024; Xie et al., 2024; Zhou et al., 2024), we propose **3D-MolT5**, a versatile T5 framework capable of understanding 3D Molecular structure to handle various 3D-dependent tasks simultaneously with text instructions. To enable LMs to comprehend 3D molecular structures, the crucial innovation is that we introduce a 3D molecular tokenization method based on the Extended 3D Fingerprint (E3FP) algorithm (Axen et al., 2017). Specifically, E3FP tokenizes the 3D molecular structure into discrete 3D tokens, with each token encapsulating the 3D information of a substructure centered around a specific atom. Since most 1D SELFIES tokens (*e.g.*, [C] and [O]) represent specific atoms, the tokens from both 1D and 3D modalities can be directly aligned at the atomic level. The embeddings of the same atom in both 1D and 3D tokens are then summed to form the final joint representation. In this way, it enables effective learning of molecular information by leveraging both sequence and structure tokens for molecules, allowing the representation of 1D molecule, 3D molecule, and 1D text modalities all using discrete tokens, such that all modalities can be easily trained in LMs. Therefore, we not only remove the dependency and requirement on external molecular structure encoders but also eliminate the necessity of challenging modality alignment training.

With tokenized 1D and 3D molecules, we conduct comprehensive molecule-text pre-training on our 3D-MolT5 framework. The pre-training tasks are inspired by the "T5 objective" (Raffel et al., 2020), which employs a "recover masked spans" objective. In 3D-MolT5, we design five types of pre-training tasks: (1) *1D denoising*: Apply T5 objective to SELFIES, text, and wrapped text, where molecules mentioned in the text are replaced with SELFIES. (2) *1D + 3D joint denoising*: Apply T5 objective to the summed 1D and 3D tokens, with the target being to recover the masked 1D SELFIES tokens. (3) *3D to 1D translation*: Given the 3D molecular tokens, generate the corresponding 1D SELFIES. (4) *3D molecule to text translation*: Given the summed 1D and 3D tokens, generate its textual description. (5) *Text to 1D molecule translation*: Given the textual description, generate the corresponding 1D SELFIES. Consequently, our 3D-MolT5 pre-training allows for early and extensive interaction between different modalities in the pre-training stage, enhances representation, and better integrates information across modalities.

To verify our 3D-MolT5 framework, we conduct instruction tuning after pre-training on various molecule-text tasks, including molecular property prediction (both 3D-dependent and 3D-independent), molecule captioning (3D-dependent), and text-based molecule generation (3D-independent). The results show that both the *Specialist* (single-task tuned) and *Generalist* (multi-task tuned) versions of 3D-MolT5 achieve superior performance across these tasks. For example, on the 3D-dependent molecular property prediction task with the PubChemQC (Maho, 2015) dataset, 3D-MolT5 achieves an improvement of nearly 70% compared to 3D-MoLM (Li et al., 2023c). These results underscore the versatility and efficacy of our 3D-MolT5 in both 3D-dependent and 3D-independent molecule-text tasks.

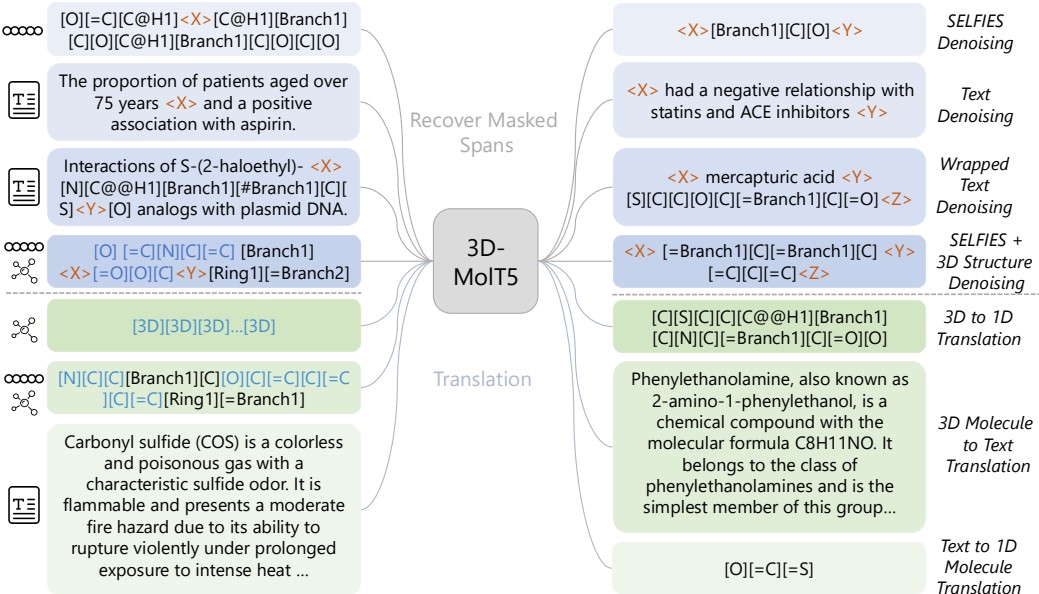

Figure 1: Overview of the 3D-MolT5 multi-task pre-training. The upper 4 tasks involve the "recover masked spans" task, where consecutive spans of the input are replaced with sentinel tokens such as <X>, <Y>, <Z>. The bottom 3 tasks are translation tasks. The input modalities are annotated with small icons. Tokens with 3D structure information are colored in blue, and [3D] refers to 3D tokens.

## 2 RELATED WORK

**Molecular Encoding.** The 1D sequence is the most widely used form of molecular encoding, typically obtained by traversing the atoms in a molecular graph in a specified order. The simplified molecular-input line-entry system (SMILES) (Weininger, 1988; Weininger et al., 1989) is the most common, while Self-Referencing Embedded Strings (SELFIES) (Krenn et al., 2020; 2022) has recently gained popularity due to its robust nature. 2D graph representations align naturally with molecular topological structures, as molecules inherently form 2D graphs with atoms serving as nodes and chemical bonds as edges (Guo et al., 2023). In contrast, 3D structures provide information about the spatial arrangement of atoms, offering valuable insights into molecular geometry and interactions. Molecular fingerprints (FPs) are also widely used, especially in molecular similarity searches and virtual screening (Cereto-Massagué et al., 2015). FPs encode critical information about molecular structure as a sequence of binary bits, which are useful for property predictions (Jeon & Kim, 2019; Wen et al., 2022). Common examples include Morgan FPs, such as extended-connectivity fingerprints (ECFPs) and functional class fingerprints (FCFPs) (Rogers & Hahn, 2010a), as well as RDKit (topological) fingerprints (Landrum et al., 2023). However, these fingerprints primarily capture 2D topological features and do not account for 3D structural patterns. Spherical extended 3D fingerprints (E3FPs) (Axen et al., 2017) effectively incorporate neighboring atoms in 3D space to encode 3D information. Our structure-aware 3D molecular vocabulary is built on E3FP (Axen et al., 2017), which is then used for our atom-centric joint representation.

**Molecule-Text Cross Modeling.** Recent advancements have integrated LMs with molecules to enhance the understanding of molecular structures and properties (Zhang et al., 2024b; Pei et al., 2024b; Taylor et al., 2022). MolT5 (Edwards et al., 2022), BioT5 (Pei et al., 2023), and BioT5+ (Pei et al., 2024a) are T5-based (Raffel et al., 2020) models that jointly trained on 1D molecular sequences and text sequences, followed by fine-tuning for tasks related to molecules. Mol-Instructions (Fang et al., 2023) and LlaSMol (Yu et al., 2024) offer instruction datasets where molecules are represented as SMILES or SELFIES for instruction tuning. Additionally, 2D molecular graphs have been utilized to infuse topological knowledge into LMs via external graph encoding modules. For instance, MoMu (Su et al., 2022), MoleculeSTM (Liu et al., 2023a), and MolFM (Luo et al., 2023) employ cross-modal contrastive learning on 2D molecular graphs and corresponding text. MolCA (Liu et al., 2023c) and MolX (Le et al., 2024) align 2D molecular space with text space through cross-modal pre-training. UniMoT (Zhang et al., 2024a) proposes a Vector Quantization-driven tokenizer to convert the 2D molecular graphs into molecule tokens, aiming at unified modeling of molecule and text. Recent endeavors have also incorporated 3D molecular information. MolBind (Xiao et al., 2024),

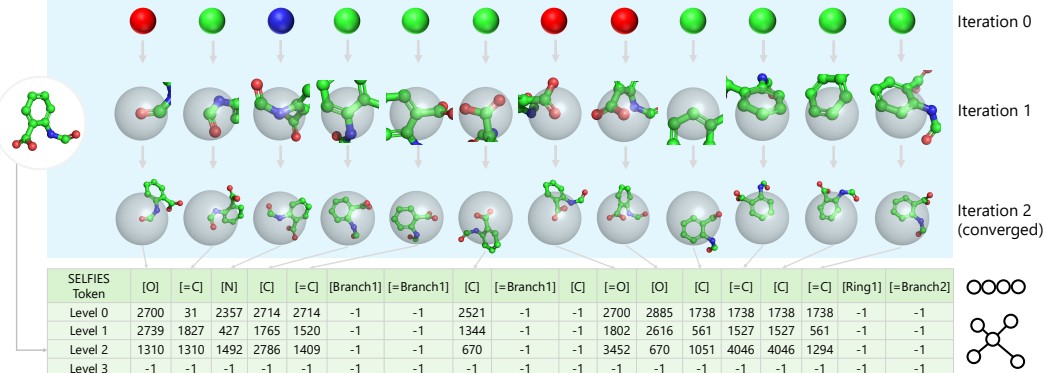

| SELFIES Token | [O] | [=C] | [N] | [C] | [=C] | [Branch1] | [=Branch1] | [C] | [=Branch1] | [C] | [=O] | [O] | [C] | [=C] | [C] | [=C] | [Ring1] | [=Branch2] |
|---|---|---|---|---|---|---|---|---|---|---|---|---|---|---|---|---|---|---|
| Level 0 | 2700 | 31 | 2357 | 2714 | 2714 | -1 | -1 | 2521 | -1 | -1 | 2700 | 2885 | 1738 | 1738 | 1738 | 1738 | -1 | -1 |
| Level 1 | 2739 | 1827 | 427 | 1765 | 1520 | -1 | -1 | 1344 | -1 | -1 | 1802 | 2616 | 561 | 1527 | 1527 | 561 | -1 | -1 |
| Level 2 | 1310 | 1310 | 1492 | 2786 | 1409 | -1 | -1 | 670 | -1 | -1 | 3452 | 670 | 1051 | 4046 | 4046 | 1294 | -1 | -1 |
| Level 3 | -1 | -1 | -1 | -1 | -1 | -1 | -1 | -1 | -1 | -1 | -1 | -1 | -1 | -1 | -1 | -1 | -1 | -1 |

Figure 2: The process of 3D molecular tokenization and alignment between 1D SELFIES tokens and 3D tokens. We choose one conformer of the 2-(Formylamino)benzoic acid (CID: 101399) as the example. At each iteration of E3FP, each atom and its neighborhood substructure is represented by a 3D token. The alignment between 1D SELFIES tokens and 3D tokens is shown at the bottom table.

for example, employs contrastive learning to align the 2D graph encoder, 3D structure encoder, and language encoder, demonstrating strong performance in cross-modal retrieval tasks. In line with the BLIP2 (Li et al., 2023b) paradigm, 3D-MoLM (Li et al., 2023c) equips the LM with an external Uni-Mol encoder (Zhou et al., 2023) and curates the 3D-MoIT dataset for 3D molecule-text instruction tuning. 3D-MoLM combines 1D SMILES and 3D molecular representations for 3D molecule-to-text interpretation. Nonetheless, these methods do not unify the modeling of molecular sequences, molecular structures, and text sequences, as the 3D molecules are encoded by an external module, posing challenges in attaining a comprehensive integration of multimodal molecular information.

## 3 METHODS

The overview of 3D-MolT5 is shown in Figure 1. We first introduce the sequence representation of molecules in Section 3.1, as it is the preliminary knowledge for our molecular tokenization. In Section 3.2, we present the 3D structure-aware E3FP fingerprint and how we adopt it in 3D-MolT5. We then introduce our 3D molecular tokenization and the integration with 1D tokenization in Section 3.3. Lastly, we present our multi-task pre-training framework in Section 3.4.

### 3.1 1D SEQUENCE REPRESENTATION

The 1D sequence representation for molecules lays the groundwork for our molecular tokenization (Section 3.3), hence we give the necessary descriptions here. In this work, for a given molecule $M$, we use SELFIES (Krenn et al., 2020) as its sequence representation, offering enhanced robustness and validity compared to SMILES (Weininger, 1988; Weininger et al., 1989). In SELFIES, each token, generally denoting an atom group (like [C] and [=N]) or structure directive (like [Ring1] and [=Branch1]), is enclosed within brackets, facilitating straightforward tokenization based on these demarcations. Therefore, the 1D SELFIES sequence can be represented as $S = \{s_i\}_{i=0}^{m-1}$, where $s_i$ is a SELFIES token and $m$ denotes sequence length[1]. To ensure that each molecule has a unique SELFIES, and thus a unique atom flatten order, we employ the canonical form of SELFIES.

### 3.2 3D STRUCTURE-AWARE FINGERPRINT

In our 3D tokenization, the focus is on transforming the continuous 3D molecular structure into discrete tokens. To achieve this, we leverage the 3D molecular fingerprint E3FP (Axen et al., 2017), which efficiently converts 3D structures into discrete identifiers, enabling a tokenized representation of spatial molecular information. For a molecule $M$ composed of $n$ atoms and one of its 3D conformers, the E3FP algorithm generates a 3D fingerprint $F$, represented as a bit vector, with $|F|$ indicating its length. We transform the atoms of $M$ into a canonical atom sequence $A = \{a_i\}_{i=0}^{n-1}$, where $a_i$ represents one of the heavy atoms. For each atom $a_i$, with $k$ representing the number of iterations for the E3FP algorithm, we can derive its 3D token $\boldsymbol{d}_i$, composed of $k + 1$ non-negative identifiers. This process, illustrated in Figure 2 and Algorithm 1 (some details and special cases

---

[1]Typically, $m$ is large than the number of atoms $n$, as in SELFIES there are some structure directive tokens.

are omitted and provided in the Appendix A), encompasses several steps as follows. (1) *Structure representation initialization*. At iteration 0, we establish a set of atomic invariants $A_i$ for each atom $a_i$ within $M$, including attributes such as atomic number, the number of immediate neighbors, and whether the atom is part of a ring, among others. These atomic invariants are hashed into identifier $\hat{d}_{i,0}$ by MurmurHash3 (Appleby, 2016) algorithm to create unique identifiers for each atom $a_i$. (2) *Iterative spherical shell expansion*. In each iteration $j$, we expand the radius of the spherical shells centered on each atom $a_i$ by a radius multiplier $r$, including all atoms within the expanded radius $R = r \cdot j$. The connectivity identifier, $c_k^i$, derived from *Connectivity*, and the stereochemical identifier, $s_k^i$, obtained from *Stereochemistry*, of these neighboring atoms are encoded relative to the central atom, incorporating both bonded and non-bonded interactions. These identifiers allow E3FP to capture the 3D molecular structure, including relative atomic orientations and distances, which are absent in 2D representations. A detailed explanation of *Connectivity* and *Stereochemistry* is provided in Appendix A.1. The current iteration number $j$, the identifier of $a_i$ from the previous iteration, and the neighbors' information ($c_k^i$, $\hat{d}_{k,j-1}$ $s_k^i$) are combined and hashed into the identifier $\hat{d}_{i,j}$. Each iteration produces a new layer of structural information, continuing until either a predefined maximum number of iterations $k$ is reached or all atoms in $M$ are included in the current shell.

(3) *Folding*. For each atom $a_i$, we aggregate its substructure information from each layer by concatenating the hashed identifiers, resulting in the vector $\hat{\boldsymbol{d}}_i = [\hat{d}_{i,0}, \hat{d}_{i,1}, \ldots, \hat{d}_{i,k}]$. Each 32-bit element of $\hat{\boldsymbol{d}}_i$ is then "folded" down to $|F|$ bit by applying the modulo operation, mathematically represented as $\boldsymbol{d}_i = \hat{\boldsymbol{d}}_i \mod |F|$. In 3D-MolT5, we do not use the E3FP $F$ directly but instead employ the 3D tokens $\boldsymbol{d}_i$ for each atom $a_i$, which are then integrated with the 1D sequence. More details and analysis about the E3FP algorithm, including connectivity and stereochemistry encoding, SE(3)-invariance, time and space complexity, the hyperparameter settings, special cases, a specific example for better illustration, information loss brought by discrete representation, are introduced in Appendix A.

### 3.3 MOLECULAR TOKENIZATION

After obtaining the 1D SELFIES sequence $S = \{s_i\}_{i=0}^{m-1}$ and the sequence of 3D tokens $\mathbf{D} = \{\boldsymbol{d}_i\}_{i=0}^{n-1}$, we combine them to form the final 1D + 3D joint representation. As depicted in Figure 2, for each molecule, most 1D SELFIES tokens $s_i$ uniquely represent an atom $a_i$. Similarly, each 3D token $\boldsymbol{d}_i$, a $k + 1$ dimensional vector of non-negative identifiers, also uniquely corresponds to an atom $a_i$. Thus, tokens from both 1D and 3D modalities can be aligned at the atomic level. Based on this alignment, we construct the 1D + 3D joint representation, capturing both the chemical sequence and the spatial configuration of the molecule.

---

**Algorithm 1** E3FP Algorithm. $\cup$ represents the concatenation operation.

1: **Input:** Molecule $M$, maximum iteration number $k$, shell radius multiplier $r$
2: Initialize $\mathbf{D} \leftarrow \{\}$
3: // Step 1: Structure Representation Initialization
4: **for** each atom $a_i$ in $M$ **do**
5:     Initialize atomic invariants $A_i$
6:     $\hat{d}_{i,0} \leftarrow$ *MurmurHash3*$(A_i)$
7:     $\hat{\boldsymbol{d}_i} \leftarrow [\hat{d}_{i,0}]$
8: **end for**
9: // Step 2: Iterative Spherical Shell Expansion
10: **for** iteration $j = 1$ to $k$ **do**
11:     $R \leftarrow r \cdot j$
12:     **for** each atom $a_i$ in $M$ **do**
13:         $L \leftarrow [j, \hat{d}_{i,j-1}]$
14:         **for** each neighbor $a_k$ within radius $R$ **do**
15:             $c_k^i \leftarrow$ *Connectivity*$(a_k, a_i)$
16:             $s_k^i \leftarrow$ *Stereochemistry*$(a_k, a_i)$
17:             $L \leftarrow L \cup [c_k^i, \hat{d}_{k,j-1}, s_k^i]$
18:         **end for**
19:         $\hat{d}_{i,j} \leftarrow$ *MurmurHash3*$(L)$
20:         $\hat{\boldsymbol{d}_i} \leftarrow \hat{\boldsymbol{d}_i} \cup [\hat{d}_{i,j}]$
21:     **end for**
22: **end for**
23: // Step 3: Folding
24: **for** each atom $a_i$ in $M$ **do**
25:     $\boldsymbol{d_i} \leftarrow \hat{\boldsymbol{d}_i} \mod |F|$
26:     $\mathbf{D} \leftarrow \mathbf{D} \cup \boldsymbol{d_i}$
27: **end for**
28: **Output:** 3D structure identifier matrix $\mathbf{D}$

---

We define the 1D embedding $\mathbf{E}_{1D}$ for the 1D SELFIES tokens and the 3D embedding $\mathbf{E}_{3D}$ for the 3D tokens. The 3D embedding $\mathbf{E}_{3D}$ is directly indexed by the 3D tokens, as each component of $\boldsymbol{d}_i$ is a non-negative integer. For each molecule, each SELFIES token $s_i$ is mapped to its corresponding embedding vector $\mathbf{E}_{1D}(s_i) \in \mathbb{R}^H$, where $H$ denotes the hidden dimension. For the 3D token embeddings, we map each component $\boldsymbol{d}_{i,j}$ in $\boldsymbol{d}_i$ to the vector $\mathbf{E}_{3D}(\boldsymbol{d}_{i,j}) \in \mathbb{R}^H$. These $k + 1$ vectors $\mathbf{E}_{3D}(\boldsymbol{d}_{i,j})$ are then averaged to compute the 3D embedding for token $\boldsymbol{d}_i$, given by $\mathbf{E}_{3D}(\boldsymbol{d}_i) = (1/k + 1) \sum_{j=0}^{k} \mathbf{E}_{3D}(\boldsymbol{d}_{i,j})$. The final joint representation $\mathbf{E}$ for each

token is determined based on the available information: if only 1D information is present, $\mathbf{E} = \mathbf{E}_{1D}$; if only 3D information is available, $\mathbf{E} = \mathbf{E}_{3D}$; and if both 1D and 3D information are present, then $\mathbf{E} = \frac{1}{2}\mathbf{E}_{1D} + \frac{1}{2}\mathbf{E}_{3D}$. This combination captures both the sequential and spatial information of the molecule, producing a comprehensive representation suitable for various downstream tasks.

## 3.4 PRE-TRAINING

Using the molecular tokenization approach described above, now we can train LMs on text sequences, molecular sequences, and also molecular structures all in tokens. Our LM backbone is T5 (Raffel et al., 2020), a transformer-based encoder-decoder architecture, which serves as the foundation for 3D-MolT5. Detailed model configurations are provided in Appendix D. For pre-training, we design two categories of tasks within a multi-task framework: (1) self-supervised denoising tasks aimed at recovering masked spans, and (2) translation tasks between different modalities to further enhance the model's capability (see ablation study in Section 5). Additional details regarding pre-training, including loss functions, configurations, and datasets, are provided in Appendix E.

**Denoising Tasks.** The denoising pre-training tasks are divided into two categories based on input modalities: (1) *1D denoising*, which includes denoising on SELFIES, text, and "wrapped" text. For SELFIES, we random sample 50M molecules from the PubChem (Kim et al., 2019) database and represent them as canonical SELFIES. For text, we use both the C4 (Raffel et al., 2020) dataset from the general domain and full articles from PubMed Central (Canese & Weis, 2013; White, 2020) in the biomedical domain. The concept of "wrapped" text is adapted from MolXPT (Liu et al., 2023b), where molecules mentioned in the text are replaced with their SELFIES. As demonstrated in Liu et al. (2023b); Pei et al. (2023; 2024a), training on such "wrapped" text is beneficial, as the context is rich with molecular descriptions. We follow the same pipeline as MolXPT to detect molecules mentioned in the text using BERN2 (Sung et al., 2022) on the PubMed abstracts, appending them with their corresponding SELFIES. (2) *1D + 3D joint denoising*, which involves denoising on the combined 1D SELFIES tokens and 3D tokens, aiming at recovering the 1D SELFIES tokens[2]. We use the PCQM4Mv2 dataset from the OGB Large Scale Challenge (Hu et al., 2021) for this task, which includes 3.37M DFT-calculated (Geerlings et al., 2003) 3D molecular structures.

**Translation Tasks.** We incorporate three translation tasks simultaneously to further bridge different modalities: (1) *3D to 1D translation*. We use the same PCQM4Mv2 dataset as in the *1D + 3D* denoising. The input is the sequence of 3D token representations $\mathbf{E}_{3D}$ of the molecule, and the output is the corresponding 1D SELFIES. (2) *3D molecule to text translation*. We use the pre-training split of the PubChem dataset collected by 3D-MoLM (Li et al., 2023c), which contains 298K 3D molecule-text pairs from the PubChem (Kim et al., 2019) database. The input is the combined 1D and 3D tokens of the molecule, and the output is the corresponding textual description. (3) *Text to 1D molecule translation*. We use the same data as (2), but the input is the textual description of the molecule, and the output is the corresponding 1D SELFIES.

## 4 EXPERIMENTS

We evaluate 3D-MolT5 on three types of text-based molecule-related downstream tasks: (1) molecular property prediction, including both 3D-independent properties (*e.g.*, molecular weight, LogP) and 3D-dependent properties (*e.g.*, HOMO-LUMO gap); (2) 3D molecule captioning; (3) text-based molecule generation. In Appendix B, we further validate the effectiveness of 3D-MolT5 across additional tasks and benchmarks. All downstream data is formatted as instructions, with tasks framed as conditional text or molecule generation based on the input instructions. The training objective remains the standard cross-entropy loss, consistent with pre-training.

Following 3D-MoLM (Li et al., 2023c) and Mol-Instructions (Fang et al., 2023), we present results for two variants of 3D-MolT5: *Specialist*, fine-tuned for a specific task; and *Generalist*, fine-tuned in a multi-task setup. To ensure fair comparisons, we use the same multi-task setup as the baseline models. More details on the downstream datasets, *Generalist* settings, and baseline methods are provided in Appendix F.

---

[2]In our work, since each 3D molecular token for an atom contains $k + 1$ different components, we do not attempt to predict the 3D molecular tokens for denoising tasks, as is also the case for the translation tasks.

Table 1: MAE results of computed property prediction tasks on PubChem (Kim et al., 2019) and PubChemQC (Maho, 2015) datasets. The valid answer rate is also reported as LMs may fail to generate valid numerical responses. 3D-dependent properties are colored in blue. † refers to a variant of 3D-MoLM (Li et al., 2023c) that is initially pre-trained on the original PubChem text without GPT-3.5 enrichment. * represents no fine-tuning.

| DATASET | PUBCHEM | | | | PUBCHEMQC | | |
|---|---|---|---|---|---|---|---|
| MODEL | WEIGHT (G/MOL) | LOGP | TPSA (Å²) | COMPLEXITY | HOMO (EV) | LUMO (EV) | H–L GAP (EV) |
| *Non-LM* | | | | | | | |
| Uni-Mol | 20.35 | 0.59 | 13.48 | 57.24 | 0.32 | 0.35 | 0.21 |
| *Specialist* | | | | | | | |
| Llama2-7B | 22.10 (96%) | 1.45 (95%) | 15.87 (92%) | 69.74 (93%) | 1.24 (96%) | 1.04 (95%) | 0.88 (92%) |
| 2D-MoLM | 21.48 (94%) | 0.88 (96%) | 13.52 (92%) | 55.74 (94%) | 0.92 (98%) | 0.80 (96%) | 0.67 (93%) |
| 3D-MoLM† | 16.18 (96%) | 0.95 (96%) | 10.26 (94%) | 49.15 (95%) | 0.45 (98%) | 0.36 (96%) | 0.41 (94%) |
| 3D-MoLM | 14.79 (95%) | 0.66 (97%) | 9.71 (93%) | 44.85 (94%) | 0.26 (97%) | 0.25 (94%) | 0.28 (94%) |
| **3D-MolT5** | **12.30** (100%) | **0.44** (100%) | **3.93** (100%) | **29.51** (100%) | **0.08** (100%) | **0.08** (100%) | **0.08** (100%) |
| *Generalist* | | | | | | | |
| Llama2-7B* | 42.18 (82%) | 2.10 (85%) | 27.11 (84%) | 121.87 (76%) | 2.87 (70%) | 1.89 (71%) | 1.86 (70%) |
| Llama2-7B | 27.42 (92%) | 1.78 (93%) | 17.07 (90%) | 78.16 (92%) | 1.89 (90%) | 1.26 (90%) | 1.25 (91%) |
| 2D-MoLM | 20.80 (92%) | 1.36 (94%) | 12.47 (89%) | 52.70 (91%) | 1.52 (93%) | 1.13 (90%) | 1.09 (88%) |
| 3D-MoLM† | 19.54 (93%) | 0.92 (92%) | 11.14 (92%) | 54.68 (90%) | 0.65 (94%) | 0.41 (92%) | 0.55 (89%) |
| 3D-MoLM | 16.58 (92%) | 0.78 (95%) | 10.90 (90%) | 45.49 (89%) | 0.35 (95%) | 0.36 (93%) | 0.32 (90%) |
| **3D-MolT5** | **14.54** (100%) | **0.61** (100%) | **6.37** (100%) | **37.59** (100%) | **0.11** (100%) | **0.11** (100%) | **0.11** (100%) |

## 4.1 MOLECULAR PROPERTY PREDICTION

Following 3D-MoLM (Li et al., 2023c), we assess 3D-MolT5 on two types of molecular property prediction tasks: (1) *Computed property prediction*: We focus on the MAE performance by extracting the predicted numerical value of the property from the generated text. (2) *Descriptive property prediction*: We evaluate text similarity metrics between the predicted text and the ground truth.

### 4.1.1 COMPUTED PROPERTY PREDICTION

**Setup.** We use three datasets to evaluate the performance of 3D-MolT5 on computed property prediction task: QM9 (Ruddigkeit et al., 2012; Fang et al., 2023), PubChemQC (Maho, 2015; Xu et al., 2021), and PubChem (Kim et al., 2019). The QM9 (Ruddigkeit et al., 2012; Fang et al., 2023) dataset contains over 130,000 molecules with ground-state 3D structures obtained through DFT computations (Geerlings et al., 2003), each molecule having fewer than nine heavy atoms. This dataset is widely used for quantum property prediction, including HOMO, LUMO, and HOMO-LUMO gap (H-L gap). The PubChemQC (Maho, 2015; Xu et al., 2021) dataset is larger in scale, containing 3.37M molecules with more heavy atoms, along with their DFT-calculated (Geerlings et al., 2003) 3D structures. We use the same quantum properties as QM9 for PubChemQC. The PubChem (Kim et al., 2019) database also provides various molecular properties. Following 3D-MoLM (Li et al., 2023c), we select four properties that can be inferred from 1D or 2D molecular information.

For the QM9 dataset, we use instruction data from Mol-Instructions (Fang et al., 2023). For PubChemQC and PubChem datasets, we use instruction data constructed by 3D-MoLM (Li et al., 2023c).

**Baselines & Evaluation.** We compare 3D-MolT5 against three types of baseline models, categorized by input modalities. For 1D sequence-based models, we include Llama2-7B (Touvron et al., 2023), Vicuna (Chiang et al., 2023), Mol-Instructions (Chiang et al., 2023), and BioT5+ (Pei et al., 2024a). For 2D graph-based models, we incorporate 2D-MoLM (Li et al., 2023c) and InstructMol (Cao et al., 2023). For the 3D structure-based model, we compare against Uni-Mol (Zhou et al., 2023) and 3D-MoLM (Li et al., 2023c). Note that models based on 2D graphs or 3D structures can also accept 1D sequences as input (Li et al., 2023c; Cao et al., 2023).

**Results.** The computed property results for the PubChem and PubChemQC datasets are presented in Table 1, and for QM9 in Table 2. Key findings from the results include: (1) 3D-MolT5 outperforms all baseline methods on 3D-independent properties in the PubChem dataset (Kim et al., 2019). For molecular hydrophobicity (LogP), which depends on 1D and 2D features such as functional groups, molecular connectivity, and topology, 3D-MolT5 consistently surpasses the LMs trained on 1D, 2D, and 3D molecular information. (2) 3D-MolT5 exhibits substantial improvements in predicting 3D-dependent properties. For energy properties including HOMO, LUMO, and HOMO-LUMO gap, which are primarily determined by 3D molecular structures, 3D-MolT5 shows significant performance enhancements. For the *Specialist* version on the PubChemQC (Maho, 2015) dataset,

Table 2: MAE results on computed property prediction tasks on QM9 (Ruddigkeit et al., 2012) dataset. * means direct inference without further fine-tuning.

| MODEL | HOMO (Ha) | LUMO (Ha) | H-L GAP (Ha) | AVG (Ha) |
|---|---|---|---|---|
| *Generalist* | | | | |
| Llama2-7B* (5-shot) | 0.7367 | 0.8641 | 0.5152 | 0.7510 |
| Vicuna-13B* (5-shot) | 0.7135 | 3.6807 | 1.5407 | 1.9783 |
| Mol-Instructions | 0.0210 | 0.0210 | 0.0203 | 0.0210 |
| BioT5+ | 0.0022 | 0.0024 | 0.0028 | 0.0025 |
| InstructMol-G | 0.0060 | 0.0070 | 0.0082 | 0.0070 |
| InstructMol-GS | 0.0048 | 0.0050 | 0.0061 | 0.0050 |
| **3D-MolT5** | **0.0017** | **0.0016** | **0.0016** | **0.0017** |

Table 3: Results of the descriptive property prediction task on PubChem (Kim et al., 2019) dataset. † refers to a variant of 3D-MoLM (Li et al., 2023c) that is initially pre-trained on the original PubChem text without GPT-3.5 enrichment. * means direct inference without further fine-tuning.

| MODEL | BLEU-2 | BLEU-4 | ROUGE-1 | ROUGE-2 | ROUGE-L | METEOR |
|---|---|---|---|---|---|---|
| *Specialist* | | | | | | |
| Llama2-7B | 28.15 | 23.24 | 35.14 | 22.08 | 30.41 | 46.87 |
| 2D-MoLM | 30.84 | 25.09 | 38.46 | 24.22 | 33.04 | 50.92 |
| 3D-MoLM† | 30.33 | 24.47 | 38.48 | 23.93 | 32.98 | 51.33 |
| 3D-MoLM | 32.00 | 26.13 | 40.13 | 25.55 | 34.64 | 52.15 |
| **3D-MolT5** | **51.24** | **43.06** | **56.06** | **40.79** | **52.48** | **56.97** |
| *Generalist* | | | | | | |
| Llama2-7B* | 25.22 | 21.16 | 31.48 | 19.21 | 25.22 | 43.17 |
| Llama2-7B | 27.68 | 22.81 | 34.73 | 21.55 | 29.91 | 46.39 |
| 2D-MoLM | 30.23 | 24.57 | 37.85 | 22.95 | 32.28 | 50.08 |
| 3D-MoLM† | 29.92 | 24.44 | 38.62 | 22.83 | 32.30 | 50.81 |
| 3D-MoLM | 31.81 | 26.08 | 40.13 | 25.87 | 34.99 | 51.93 |
| **3D-MolT5** | **49.84** | **41.42** | **54.23** | **38.45** | **50.39** | **54.76** |

the improvements over the previous SOTA methods (Zhou et al., 2023; Li et al., 2023c) are 0.18 eV, 0.17 eV, and 0.13 eV, respectively. For the *Generalist* version on the QM9 (Ruddigkeit et al., 2012) dataset, the average improvement is 0.0008 Ha. (3) Compared to Uni-Mol (Zhou et al., 2023), 3D-MolT5 exhibits consistent improvements. Uni-Mol (Zhou et al., 2023) is specially designed for 3D molecular representation learning and pre-trained on large-scale (209M) 3D molecular data. The superiority of 3D-MolT5 demonstrates the benefit of unified 3D molecule-text modeling. By integrating structural knowledge from molecules with contextual knowledge from biological literature through comprehensive pre-training, 3D-MolT5 enhances its generalization to molecular property prediction tasks. (4) 3D-MolT5 also continuously improves upon 3D-MoLM (Li et al., 2023c), which integrates the Uni-Mol (Zhou et al., 2023) molecular encoder with Llama2-7B (Touvron et al., 2023) as the language decoder through a projector for 3D molecule-text interpretation. The superiority of 3D-MolT5 can be attributed to the extensive interaction between 3D molecular structure and text during multi-task pre-training and our joint tokenization, which significantly improves 3D-MolT5's ability to handle complex fine-grained 3D molecular structures and enhance cross-modal understanding. (5) On PubChemQC and PubChem datasets, the *Generalist* version of 3D-MolT5 also outperforms all baselines, though it slightly underperforms compared to the *Specialist*, likely due to task conflicts in multi-task training. These results demonstrate 3D-MolT5's ability to handle multiple tasks concurrently as a *Generalist*.

### 4.1.2 DESCRIPTIVE PROPERTY PREDICTION

**Setup.** For the descriptive property prediction task, we evaluate the performance of 3D-MolT5 on the PubChem dataset, as constructed by 3D-MoLM (Li et al., 2023c). Unlike computed property prediction, this task involves generating natural language descriptions of molecular 1D, 2D, and 3D properties, necessitating accurate and contextually relevant textual output.

**Baselines & Evaluation.** We compare 3D-MolT5 against the same baselines used for the Pub-ChemQC (Maho, 2015) and PubChem (Kim et al., 2019) datasets as described in Section 4.1.1. Following MolT5 (Edwards et al., 2022), we employ widely used text generation metrics, including BLEU (Papineni et al., 2002), ROUGE (Lin, 2004), and METEOR (Banerjee & Lavie, 2005), to assess the similarity between the generated property descriptions and the ground truth.

**Results.** The results for the PubChem dataset are shown in Table 3. From the table, we have several observations: (1) 3D-MolT5 outperforms all baseline methods. The *Specialist* version

Table 4: Results for the 3D molecule captioning task on PubChem (Kim et al., 2019) dataset. † refers to a variant of 3D-MoLM (Li et al., 2023c) that is initially pre-trained on the original PubChem text without GPT-3.5 enrichment.

| MODEL | BLEU-2 | BLEU-4 | ROUGE-1 | ROUGE-2 | ROUGE-L | METEOR |
|---|---|---|---|---|---|---|
| *Specialist* | | | | | | |
| MolT5-Large | 25.87 | 17.28 | 34.07 | 16.42 | 23.41 | 28.04 |
| MoMu-Large | 26.34 | 18.01 | 34.75 | 16.86 | 24.76 | 28.73 |
| 3D-MoLM† | 29.82 | 22.39 | 37.23 | 22.49 | 31.07 | 32.69 |
| 3D-MoLM | 30.32 | 22.52 | 36.84 | 22.32 | 31.23 | 33.06 |
| UniMoT | 31.30 | 23.80 | 37.50 | 23.70 | 33.60 | 34.80 |
| MolX | 31.40 | 24.25 | 44.20 | 28.96 | 38.76 | 39.55 |
| **3D-MolT5** | **42.05** | **34.16** | **48.13** | **33.20** | **42.33** | **44.69** |
| *Generalist* | | | | | | |
| Llama2-7B | 27.01 | 20.94 | 35.76 | 20.68 | 28.88 | 32.11 |
| 2D-MoLM | 27.15 | 21.19 | 36.02 | 20.76 | 29.12 | 32.28 |
| 3D-MoLM† | 29.25 | 22.07 | 36.48 | 21.80 | 30.95 | 33.12 |
| 3D-MoLM | 28.95 | 21.63 | 36.51 | 21.26 | 30.02 | 33.55 |
| **3D-MolT5** | **43.11** | **34.98** | **48.69** | **33.54** | **42.72** | **44.82** |

shows significant improvements, with BLEU-2 and ROUGE-L scores increasing by 19.24 and 17.84, respectively. The *Generalist* version also shows substantial gains, with BLEU-2 and ROUGE-L improvements of 18.03 and 15.4 points, respectively. (2) The *Generalist* version of 3D-MolT5 surpasses all baselines but slightly underperforms compared to the *Specialist* version.

## 4.2 3D MOLECULE CAPTIONING AND TEXT-BASED MOLECULE GENERATION

Despite the molecular property prediction tasks, we also evaluate our 3D-MolT5 on 3D molecule captioning and text-based molecule generation tasks.

### 4.2.1 3D MOLECULE CAPTIONING

**Setup.** We use PubChem (Li et al., 2023c) dataset for 3D molecule captioning to evaluate 3D-MolT5's ability to understand the 3D molecular structure. This dataset contains approximately 15,000 3D molecular structure-text pairs sourced from the PubChem database (Kim et al., 2019). Specifically, the molecular captions include both molecular names and 3D-related descriptions to assess the model's capability in name prediction (Favre & Powell, 2013) and description prediction (Edwards et al., 2022), offering a more comprehensive evaluation compared to the descriptive properties.

**Baselines & Evaluation.** The compared baselines are classified into three categories based on input modalities. For 1D sequence-based models, we include MolT5 (Edwards et al., 2022) and Llama2-7B (Touvron et al., 2023). For 2D graph-based models, we compare against MoMu (Su et al., 2022), 2D-MoLM (Li et al., 2023c), UniMoT (Zhang et al., 2024a), and MolX (Le et al., 2024). For 3D structure-based models, we incorporate 3D-MoLM (Li et al., 2023c). Note that the 2D graph or 3D structure-based models may also take a 1D sequence as input simultaneously. The evaluation metrics remain consistent with those in Section 4.1.2.

**Results.** Table 4 shows the results for the 3D molecule captioning task. 3D-MolT5 demonstrates superior results, surpassing all baseline methods. Compared to 3D-MoLM (Li et al., 2023c), 3D-MolT5 achieves an improvement of approximately 11 points in ROUGE-L and METEOR scores for both the *Specialist* and *Generalist* versions. Furthermore, 3D-MolT5 also exceeds baselines with 1D and 2D information, highlighting the importance of 3D structure information for molecule understanding and captioning. This further validates the efficacy of our unified pre-training on 1D SELFIES, 3D structure, and text with 3D molecular tokenization.

### 4.2.2 TEXT-BASED MOLECULE GENERATION

**Setup.** To further demonstrate the capability of 3D-MolT5, we also evaluate its performance on the text-based molecule generation task. We use the CheBI-20 (Edwards et al., 2022) dataset, which is widely used for this task (Edwards et al., 2022; Luo et al., 2023; Liu et al., 2024; 2023b; Pei et al., 2023). The input for this task is the textual description of the molecule, and the target is to generate a 1D molecular sequence that fits the description.

**Baselines & Evaluation.** The compared baselines include Llama2-7B (Touvron et al., 2023), GPT-3.5 (OpenAI, 2023), GPT-4 (Achiam et al., 2023), T5 (Raffel et al., 2020), MolT5 (Edwards et al.,

Table 5: Results on text-guided molecule generation task on ChEBI-20 (Edwards et al., 2022) dataset.

| MODEL | BLEU↑ | EXACT↑ | LEVENSHTEIN↓ | MACCS FTS↑ | RDK FTS↑ | MORGAN FTS↑ | FCD↓ | TEXT2MOL↑ | VALIDITY↑ |
|---|---|---|---|---|---|---|---|---|---|
| *Generalist* | | | | | | | | | |
| Llama2-7B (0-shot) | 0.104 | 0.000 | 84.18 | 0.243 | 0.119 | 0.089 | 42.01 | 0.148 | 0.631 |
| Llama2-7B (2-shot) | 0.693 | 0.022 | 36.77 | 0.808 | 0.717 | 0.609 | 4.90 | 0.149 | 0.761 |
| GPT-3.5-turbo (0-shot) | 0.489 | 0.019 | 52.13 | 0.705 | 0.462 | 0.367 | 2.05 | 0.479 | 0.802 |
| GPT-3.5-turbo (10-shot) | 0.790 | 0.139 | 24.910 | 0.847 | 0.708 | 0.624 | 0.57 | 0.571 | 0.887 |
| GPT-4-0314 (10-shot) | 0.857 | 0.280 | 17.14 | **0.903** | 0.805 | 0.739 | **0.41** | **0.593** | 0.899 |
| *Specialist* | | | | | | | | | |
| T5-base | 0.762 | 0.069 | 24.950 | 0.731 | 0.605 | 0.545 | 2.48 | 0.499 | 0.660 |
| T5-large | 0.854 | 0.279 | 16.721 | 0.823 | 0.731 | 0.670 | 1.22 | 0.552 | 0.902 |
| MolT5-base | 0.769 | 0.081 | 24.458 | 0.721 | 0.588 | 0.529 | 2.18 | 0.496 | 0.772 |
| MolT5-large | 0.854 | 0.311 | 16.071 | 0.834 | 0.746 | 0.684 | 1.20 | 0.554 | 0.905 |
| MoMu-base | 0.815 | 0.183 | 20.520 | 0.847 | 0.737 | 0.678 | - | 0.580 | 0.863 |
| MolFM-base | 0.822 | 0.210 | 19.445 | 0.854 | 0.758 | 0.697 | - | 0.583 | 0.892 |
| GIT-Mol | 0.756 | 0.051 | 26.315 | 0.738 | 0.582 | 0.519 | - | - | 0.928 |
| MolXPT | - | 0.215 | - | 0.859 | 0.757 | 0.667 | 0.45 | 0.578 | 0.983 |
| BioT5 | **0.867** | 0.413 | 15.097 | 0.886 | 0.801 | 0.734 | 0.43 | 0.576 | **1.000** |
| **3D-MolT5** | 0.849 | **0.487** | **10.527** | 0.884 | **0.806** | **0.744** | **0.41** | 0.574 | **1.000** |

2022), MoMu (Su et al., 2022), MolFM (Luo et al., 2023), GIT-Mol (Liu et al., 2024), MolXPT (Liu et al., 2023b), and BioT5 (Pei et al., 2023). Following (Edwards et al., 2022), the evaluation metrics include BLEU (Papineni et al., 2002), exact match score, levenshtein distance, fingerprint similarity (Durant et al., 2002; Landrum et al., 2023; Rogers & Hahn, 2010b), FCD score (Preuer et al., 2018), text2mol (Edwards et al., 2021) score, and validity.

**Results.** The results are presented in Table 5. Our 3D-MolT5 outperforms all compared baselines across most metrics. Notably, 3D-MolT5 achieves an exact match score of 0.487, indicating that nearly 50% of the generated molecules exactly match the ground truth molecules. These results underscore that 3D-MolT5 has acquired comprehensive molecular knowledge during pre-training, enabling it to effectively generate accurate molecular sequences based on textual descriptions.

## 5 ABLATION STUDY

To validate the efficacy of 3D molecular tokenization and multi-task pre-training, we conduct ablation studies focused on property prediction using the PubChemQC (Maho, 2015) dataset, a task heavily reliant on 3D structural information. The ablation results are shown in Figure 3. More case studies are shown in Appendix G.

**Whether 3D input truly help?** To assess the impact of 3D input, we pre-train and fine-tune a variant of 3D-MolT5 excluding all the 3D structure information. As illustrated in Figure 3, removing 3D information leads to a performance drop on the 3D-dependent

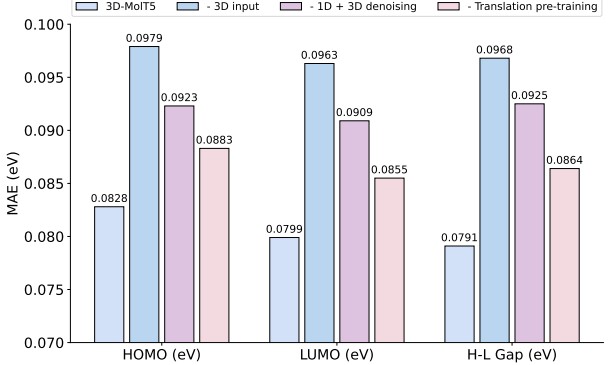

Figure 3: Ablation studies on PubChemQC (Maho, 2015) dataset. The evaluation metric is MAE.

property prediction task. For example, the MAE for the HOMO-LUMO gap increases from 0.0791 to 0.0968. This indicates that integrating 3D structure information into LMs can enhance their understanding of the molecule.

**Whether 3D-related pre-training help?** To demonstrate the efficacy of our 3D-related pre-training, we remove the 1D + 3D joint denoising task and translation tasks separately. The results in Figure 3 indicate that both of them contribute to improvements in 3D-related downstream tasks, underscoring the importance of incorporating 3D information into the pre-training process.

## 6 CONCLUSION

In this paper, we introduce 3D-MolT5, a unified framework that integrates molecular sequences, molecular structures, and text sequences, to enhance the capabilities of language models in handling various molecular tasks. By proposing a 3D molecular tokenization method, we can effectively map 3D structures to 3D tokens. The combination of 1D SELFIES tokens and 3D tokens enables a comprehensive representation of the molecule. Through extensive pre-training on 1D and 3D data and subsequent instruction tuning, 3D-MolT5 demonstrates superior performance in molecular property prediction, molecule captioning, and text-based molecule generation tasks.

ACKNOWLEDGEMENTS

We would like to thank the reviewers for their insightful comments. This work was supported by Beijing Outstanding Young Scientist Program NO. BJJWZYJH012019100020098, and the Intelligent Social Governance Platform, Major Innovation & Planning Interdisciplinary Platform for the "Double-First Class" Initiative, Renmin University of China. Qizhi Pei is supported by the Outstanding Innovative Talents Cultivation Funded Programs 2023 of Renmin University of China. Qizhi Pei is an intern at Shanghai AI Laboratory.

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

# A    MORE DETAILS ABOUT E3FP

## A.1    CONNECTIVITY AND STEREOCHEMISTRY ENCODING

In Algorithm 1, there are two functions, *Connectivity* and *Stereochemistry*, which encode the connectivity and stereochemical information between $a_k$ and $a_i$ respectively, resulting in the $c_k^i$ and $s_k^i$. The connectivity identifier $c_k^i$ ranges from 1 to 5, which represents relative atomic distance: 1 for single bond, 2 for double bond, 3 for triple bond, 4 for aromatic bonds, and 5 for no bonds. The stereochemical identifier $s_k^i$ encodes relative atomic orientation. It ranges from -5 to 5 based on their regions in a divided unit sphere, where the division is based on the x/y-axis defined by direction vectors from the center atom to its neighbors. More details can be found in the original paper (Axen et al., 2017).

## A.2    SE(3)-INVARIANCE ANALYSIS

The process of mapping 3D molecular structure to hash values in the E3FP algorithm is SE(3)-invariant. The reason is: (1) The initial identifier at iteration 0 for each atom is defined by atomic invariant features. (2) As shown in step 2 of Algorithm 1, during the iterative process of E3FP, the identifier for the current shell defined by E3FP is determined by iteration number $j$, the identifier for the same atom from the previous iteration $\hat{d}_{i,j-1}$, the neighbors' connectivity $c_k^i$ and relative orientation $s_k^i$ with respect to the center atom of the current shell, and the neighbors' identifiers $\hat{d}_{k,j-1}$ from the previous iteration. Both the initialization and iterative process are not affected by the molecule's rotation, translation, and reflection, thus preserving the SE(3)-invariance.

## A.3    COMPLEXITY ANALYSIS

In this section, we briefly analyze the complexity of E3FP algorithm.

**Time Complexity.**    The core of the E3FP algorithm involves iterating over atoms and their neighborhoods up to a fixed number of iterations $k$. At each iteration, the algorithm: (1) Draws a shell of increasing radius around each atom; (2) Identifies neighbors within the shell; (3) Generates unique identifiers for the substructures formed. For a molecule with $n$ heavy atoms, the time complexity for each iteration involves examining the neighbors within the shell. Typically, the number of neighbors is bounded by a constant $b$, due to the constraints of chemical valence. Thus, the time complexity of the fingerprinting process for each atom is $O(bk)$, and for a molecule with $n$ atoms, it becomes $O(nbk)$. Typically, $n <= 100$, $b <= 4$, and $k = 3$ in our setting.

**Space Complexity.**    As shown in Figure 2 in our paper, without considering the storage of intermediate variables, we only need to store the final 3D token indices for each atom at each iteration. Thus, for a molecule with $N$ atoms, the overall space complexity for a single molecule is $O(Nk)$. Notably, in practice, the generation of E3FP fingerprints is performed offline and can be executed rapidly through multi-processing, achieving a throughput of approximately 300 samples per second using 24 parallel processes.

## A.4    HYPERPARAMETER SETTINGS AND SPECIAL CASES

In E3FP (Axen et al., 2017), there are three key hyperparameters, the iteration number $k$, the shell radius multipler $r$, and the length of E3FP fingerprint $|F|$. The $k$ is set to 3, as our preliminary experiments indicate that three iterations are sufficient for the E3FP algorithm to converge, capturing all potentially occurring substructures for the vast majority of molecules. The $r$ is set to 1.718Å following the default setting in E3FP (Axen et al., 2017). The $|F|$ is set to 4096 rather than the default 1024 to further decrease the probability of collisions of folding.

There are two special cases where the 3D substructure identifier $\hat{d}$ will be -1, indexing a zero embedding of dimension $H$: (1) The corresponding SELFIES tokens do not correspond to atoms, like structure directive tokens: `[Ring1]` and `[=Branch1]`. (2) The E3FP algorithm is converged before the $k$ iterations.

| $a_1$ SELFIES token: [=C] | Feature Array | MurmurHash3 | 32-bit Identifier | Modulo 4096 | Folded Identifier | Iteration Number |
|---|---|---|---|---|---|---|
| | [ 6, 3, 1, 0, 0, 0 ] | | 1763934239 | | 31 | 0 |
| | [ 1, 1763934239,
1, - 615634635, 1,
2, 410692236 , -2 ] | | -915867869 | | 1827 | 1 |
| | [ 2, - 915867869 ,
1, 1990676907 , 1,
2, 815643315 , -2,
5, -2111331791 , 2,
5, 5195493 , 2 ] | | -1918577378 | | 1310 | 2 (converged) |
| | --- | | --- | | -1 | 3 |

Figure 4: Visualization of the E3FP process the second atom $a_1$ (Carbon) of the molecule with CID 101399 (same as the case in Figure 2).

The final number of embeddings for SELFIES tokens is 2944, and for 3D tokens is 4097 as there is a special zero embedding representing no 3D information.

### A.5 EXAMPLE FOR ENCODING PROCESS

To provide a clearer illustration of the E3FP process for better understanding, we present a specific case here. For the case in Figure 2, let's consider the second atom (Carbon) with SELFIES token [=C] (denoted as $a_1$ for simplicity) and its shells at each iteration. The detailed visualization of the E3FP process is shown in Figure 4.

At iteration 0, a set of atomic invariants for each atom $a_i$ is used to initialize the structure representation. For $a_1$, these invariants include the atomic number (6), the number of immediate neighbors (3), the number of bound hydrogens (1), the difference between the atomic mass and the standard atomic weight of the corresponding element (0), the atomic formal charge (0), and whether the atom is part of a ring (0). These atomic invariants are concatenated ([6, 3, 1, 0, 0, 0]) and then hashed into identifier $\hat{d}_{1,0} = 1763934239$ by MurmurHash3 (Appleby, 2016) algorithm.

At iteration 1, a spherical shell centered on $a_1$ with radius $r$ is defined. The connectivity and spatial arrangement of neighboring atoms within the shell are encoded relative to their central atom by a 2-element header list and several 3-element lists. For $a_1$, the header list is $[1, 1763934239]$, where 1 is the iteration number and 1763934239 is the identifier of the shell from the previous iteration, i.e., $\hat{d}_{1,0}$. Since there are two neighboring atoms centered on $a_1$ in this spherical shell, two 3-element lists are defined: $[1, -615634635, 1]$ for Nitrogen and $[2, 410692236, -2]$ for Oxygen. The first position of the 3-element list (e.g., 1 for Nitrogen, 2 for Oxygen) is the int connectivity identifier introduced in Appendix A.1. The second position (e.g., $-615634635$ for Nitrogen, 410692236 for Oxygen) is the identifier of the shell from the previous iteration. The third position (e.g., 1 for Nitrogen, $-2$ for Oxygen) is the stereochemical identifier encoding relative atomic orientation as illustrated in Appendix A.1. Then these lists are concatenated ($[1, 1763934239, 1, -615634635, 1, 2, 410692236, -2]$), and then hashed to identifier $\hat{d}_{1,1} = -915867869$ by MurmurHash3 (Appleby, 2016).

At iteration 2, the radius of the shell is increased to $2r$, and the process similar to iteration 1 performs again. For $a_1$, the resulting identifier at this iteration is $\hat{d}_{1,2} = -1918577378$.

In our setting, the maximum iteration number is 3. However, for this molecule example, the E3FP converges at iteration 2, as the shell at iteration 2 centered on atom $a_5$ (Carbon) has already included all atoms. The combined hash identifier for $a_1$ is $\hat{d}_1 = [\hat{d}_{1,0}, \hat{d}_{1,1}, \hat{d}_{1,2}] = [1763934239, -915867869, -1918577378]$, which is then converted to $d_1 = \hat{d}_1 \bmod |F| = [31, 1827, 1310]$. The final $d_1 = [31, 1827, 1310, -1]$, where $-1$ indicates no 3D information for iteration 3, as shown in the bottom table of Figure 2.

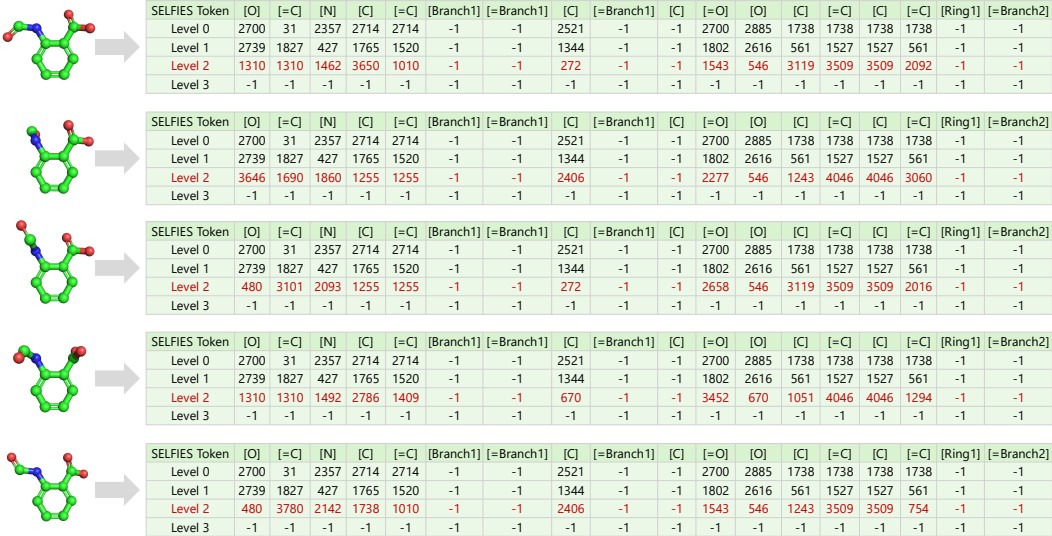

Figure 5: Visualization of 5 conformers and their corresponding 3D tokens for molecule with CID 101399 (same as Figure 2). The difference among their 3D tokens are colored in red.

### A.6 INFORMATION LOSS OF E3FP DISCRETE ENCODING

In our 3D tokenization, we use discrete tokens to represent 3D molecular substructure, which is sparser than continuous representation and may introduce information loss. We empirically demonstrate that this sparsity does not hinder the extraction of critical information for 3D structures from the following two perspectives. (1) **We show that E3FP can effectively capture subtle variations between different conformers.** We choose a molecule from PubChem and visualize its 5 conformers as shown in Figure 5. Notably, these conformers exhibit slight variations in two substituent groups on the benzene ring. We can see that different conformers possess distinct 3D tokens extracted by E3FP. Thanks to the hierarchical nature of our 3D tokenization: iteration 0 primarily captures atomic invariant features, and iteration 1 accounts for neighbors within a radius $r = 1.718\mathring{A}$. Since iterations 0 and 1 are quite local, these features are identical for atoms across different conformers. However, iteration 2 considers the neighbors within a larger radius $2r$, so it reveals subtle differences between conformers. This proves our 3D tokens can effectively capture subtle variations between different conformers. (2) **We do empirical verifications on 3D molecular understanding tasks, which is the focus of our work, to demonstrate whether the full continuous information is necessary or if the discrete token is enough for the performance effect.** Specifically, we compare a variant of our model that does not incorporate 1D SELFIES information and relies solely on 3D tokens, with Uni-Mol (Zhou et al., 2023), which employs 3D continuous information, on the H-L gap prediction task with the QM9 dataset. Our model achieves an MAE of 0.15, outperforming the Uni-Mol's 0.21. This suggests that discrete 3D tokens are sufficient for 3D understanding tasks and the information loss is not heavy. (3) **An empirical observation from recent work**. A recent study, UniMoT (Zhang et al., 2024a) introduced a Vector Quantization-driven tokenizer to convert 2D molecular graphs into sequences of molecular tokens, followed by multi-stage training, enabling joint molecule-text modeling. Their experiments on the molecule captioning task revealed that, while quantized discrete tokens exhibit slightly inferior performance compared to continuous embeddings, the performance degradation is marginal. This demonstrates that discrete representation may indeed lead to some degree of information loss, but it remains within acceptable limits.

## B ADDITIONAL DOWNSTREAM RESULTS

In Section 4, we primarily focus on 3D-related molecular tasks. However, since 3D-MolT5 can naturally adapt to tasks involving 1D and 2D molecular representations, we further evaluate its versatility on a broader range of benchmark datasets in this section. These benchmarks include

Table 6: Results for retrosynthesis task on USPTO-50k (Schneider et al., 2016) dataset (**Best**, Second Best).

| SETTING | METHOD | BLEU-2↑ | LEVENSHTEIN↓ | RDK FTS↑ | VALIDITY↑ |
|---|---|---|---|---|---|
| Infer-only | Llama-2-7B | 10.10 | 468.74 | - | 00.00 |
| | Llama-2-7B + MolX | 36.73 | 62.33 | 0.4041 | 13.71 |
| LoRA FT | Llama-2-7B | 80.37 | 16.22 | 0.6981 | 89.27 |
| | LlasMol-7B | 50.09 | 31.28 | 0.7351 | 99.65 |
| | ChemDFM-13B | 39.93 | 57.48 | 0.5380 | 14.04 |
| | Llama-2-7B + MoMu | 70.88 | 20.77 | 0.5691 | 90.53 |
| | Llama-2-7B + 2D-MoLM | 82.05 | 15.90 | 0.7126 | 91.13 |
| | Llama-2-7B + 3D-MoLM | 81.31 | 16.21 | 0.7341 | 90.31 |
| | Llama-2-7B + MoX | 82.59 | 15.74 | 0.7466 | 92.19 |
| | Llama-2-7B + MolX | 36.73 | 62.33 | 0.4041 | 13.71 |
| Full FT | Chemformer | 74.01 | 19.51 | 0.6951 | 97.14 |
| | ReactionT5-Large | 81.63 | 17.69 | 0.7400 | 97.58 |
| | **3D-MolT5** | **86.23** | **14.08** | **0.7538** | **100.00** |

Table 7: Results (AUROC) for molecule property prediction tasks on MoleculeNet (Wu et al., 2018) benchmark (**Best**, Second Best). * represents LoRA (Hu et al., 2022) tuning.

| METHOD | BACE | BBBP | HIV | Clintox | Avg |
|---|---|---|---|---|---|
| # MOLECULES | 1513 | 2039 | 41127 | 1478 | - |
| *Non-LM-based Models* | | | | | |
| GraphCL | 75.4 | 69.7 | 78.5 | 76.0 | 74.9 |
| GraphMVP-C | 81.2 | 72.4 | 77.0 | 77.5 | 77.0 |
| MGSSL | 79.7 | 70.5 | 79.5 | 80.7 | 77.6 |
| MolCLR | **89.0** | 73.8 | 80.6 | 93.2 | 84.2 |
| GEM | 85.6 | 72.4 | 80.6 | 90.1 | 82.2 |
| Uni-Mol | 85.7 | 72.9 | **80.8** | 91.9 | 82.8 |
| *LM-based Models* | | | | | |
| KV-PLM | 71.9 | 66.9 | 68.8 | 84.3 | 73.0 |
| MoMu | 76.7 | 70.5 | 75.9 | 79.9 | 75.8 |
| MolFM | 83.9 | 72.9 | 78.8 | 79.7 | 78.8 |
| Galactica-6.7B | 58.4 | 53.5 | 72.2 | 78.4 | 65.6 |
| Galactica-30B | 72.7 | 59.6 | 75.9 | 82.2 | 72.6 |
| Galactica-120B | 61.7 | 66.1 | 74.5 | 82.6 | 71.2 |
| UniMoT | 83.7 | 71.4 | 78.5 | 92.9 | 81.6 |
| **3D-MolT5** | 88.1 | **76.5** | **80.8** | **95.4** | **85.2** |

retrosynthesis on the USPTO-50k dataset (Schneider et al., 2016)(Table 6), molecular property prediction on the MoleculeNet benchmark (Wu et al., 2018)(Table 7), and tasks including forward reaction prediction, reagent prediction, and retrosynthesis on the Mol-Instructions datasets (Fang et al., 2023)(Table 8). The superior performance of 3D-MolT5 across these diverse molecular modeling tasks highlights its robustness and adaptability to various molecular benchmarks.

# C ADDITIONAL DISCUSSIONS AND FUTURE DIRECTIONS

## C.1 COMPARISON WITH DIRECT COORDINATE REPRESENTATION

Directly encoding spatial molecular data as text containing atom coordinates, as demonstrated in recent works (Zholus et al., 2024; Flam-Shepherd & Aspuru-Guzik, 2023), is indeed a simpler and more transparent approach than E3FP (Axen et al., 2017) encoding in 3D-MolT5. However, for 3D molecular understanding tasks, such as property prediction and captioning, the E3FP-based discrete token scheme offers significant advantages, which we summarize below:

(1) **Input Length and Computational Efficiency**. Representing spatial coordinates directly as text substantially increases input sequence length, especially when dealing with large molecules. This not only introduces additional computational overhead considering the quadratic complexity of the attention mechanism, but also complicates the model's learning process, as longer sequences can dilute meaningful patterns within the data. In contrast, by encoding 3D structure into 3D tokens and

Table 8: Results for chemical reaction-related tasks on Mol-Instructions (Fang et al., 2023) datasets (**Best**, Second Best). ∗ represents LoRA (Hu et al., 2022) tuning.

| MODEL | EXACT↑ | BLEU↑ | LEVENSHTEIN↓ | RDK FTS↑ | MACCS FTS↑ | MORGAN FTS↑ | VALIDITY↑ |
|---|---|---|---|---|---|---|---|
| *Reagent Prediction* | | | | | | | |
| Llama-7B | 0.000 | 0.003 | 28.040 | 0.037 | 0.001 | 0.001 | 0.001 |
| Galactica-6.7B | 0.000 | 0.141 | 30.760 | 0.036 | 0.127 | 0.051 | 0.995 |
| Text+Chem T5-223M | 0.000 | 0.225 | 49.323 | 0.039 | 0.186 | 0.052 | 0.313 |
| Mol-Instructions-7B | 0.044 | 0.224 | 23.167 | 0.237 | 0.364 | 0.213 | 1.000 |
| Llama-7B∗(LoRA) | 0.000 | 0.283 | 53.510 | 0.136 | 0.294 | 0.106 | 1.000 |
| InstructMol-G-6.9B | 0.070 | **0.890** | 24.732 | 0.469 | **0.691** | 0.426 | 1.000 |
| InstructMol-GS-6.9B | 0.129 | 0.610 | 19.664 | 0.444 | 0.539 | 0.400 | 1.000 |
| UniMoT | 0.167 | 0.728 | 14.588 | 0.549 | 0.621 | 0.507 | 1.000 |
| **3D-MolT5** | **0.277** | 0.756 | **13.173** | **0.571** | 0.646 | **0.540** | 1.000 |
| *Forward Reaction Prediction* | | | | | | | |
| Llama-7B | 0.000 | 0.020 | 42.002 | 0.001 | 0.002 | 0.001 | 0.039 |
| Galactica-6.7B | 0.000 | 0.468 | 35.021 | 0.156 | 0.257 | 0.097 | 0.946 |
| Text+Chem T5-223M | 0.239 | 0.782 | 20.413 | 0.705 | 0.789 | 0.652 | 0.762 |
| Mol-Instructions-7B | 0.045 | 0.654 | 27.262 | 0.313 | 0.509 | 0.262 | 1.000 |
| Llama-7B∗(LoRA) | 0.012 | 0.804 | 29.947 | 0.499 | 0.649 | 0.407 | 1.000 |
| InstructMol-G-6.9B | 0.153 | 0.906 | 20.155 | 0.519 | 0.717 | 0.457 | 1.000 |
| InstructMol-GS-6.9B | 0.536 | 0.967 | 10.851 | 0.776 | 0.878 | 0.741 | 1.000 |
| UniMoT | 0.611 | 0.980 | 8.297 | 0.836 | 0.911 | 0.807 | 1.000 |
| **3D-MolT5** | **0.879** | **0.994** | **2.842** | **0.955** | **0.978** | **0.942** | 1.000 |
| *Retrosynthesis* | | | | | | | |
| Llama-7B | 0.000 | 0.036 | 46.844 | 0.018 | 0.029 | 0.017 | 0.010 |
| Galactica-6.7B | 0.000 | 0.452 | 34.940 | 0.167 | 0.274 | 0.134 | 0.986 |
| Text+Chem T5-223M | 0.141 | 0.765 | 24.043 | 0.685 | 0.765 | 0.585 | 0.698 |
| Mol-Instructions-7B | 0.009 | 0.705 | 31.227 | 0.283 | 0.487 | 0.230 | 1.000 |
| Llama-7B∗(LoRA) | 0.000 | 0.283 | 53.510 | 0.136 | 0.294 | 0.106 | 1.000 |
| InstructMol-G-6.9B | 0.114 | 0.586 | 21.271 | 0.422 | 0.523 | 0.285 | 1.000 |
| InstructMol-GS-6.9B | 0.407 | 0.941 | 13.967 | 0.753 | 0.852 | 0.714 | 1.000 |
| UniMoT | 0.478 | 0.974 | 11.634 | 0.810 | 0.909 | 0.771 | 1.000 |
| **3D-MolT5** | **0.679** | **0.974** | **6.201** | **0.907** | **0.940** | **0.878** | 1.000 |

aligning 1D and 3D embeddings at the atomic level, 3D-MolT5 maintains a balanced and scalable representation while avoiding unnecessary computational complexity.

(2) **Semantic Representation of Numerical Data**. Tokenizing coordinates as text often results in a loss of numerical semantic relationships. For instance, the tokens for the numbers "123" and "124" are treated as entirely distinct, despite their numerical proximity. This lack of semantic similarity makes it challenging for the model to capture meaningful numerical relationships[3], such as proximity or continuity. In contrast, the E3FP algorithm encodes 3D molecular structures as discrete tokens based on hierarchical and spatial substructures, preserving critical spatial relationships in a form that is more interpretable and useful for the model.

(3) **Preservation of SE(3)-Invariance**. Representing spatial data directly as coordinates can struggle with preserving SE(3)-Invariance, i.e., invariance to molecular rotations, translations, and reflections. Without explicit adjustments, such representations may lead to inconsistencies in encoding the same molecule under different orientations. But E3FP is inherently SE(3)-invariant (discussed in Appendix A.2), ensuring that the discrete tokens remain consistent regardless of molecular orientation, which is crucial for tasks like 3D molecular understanding.

While 3D-MolT5 currently focuses on 3D molecular understanding tasks, we recognize the potential of extending the model to support 3D structure generation. Future work could integrate both 1D and E3FP-based 3D tokens into a unified sequence and incorporate an external decoder to reconstruct 3D structures from generated tokens, addressing both understanding and generation tasks in molecular modeling.

## C.2 SEQUENTIAL INTEGRATION OF E3FP TOKENS AND 1D SELFIES TOKENS

In this section, we explore whether the E3FP tokens could be directly concatenated with 1D SELFIES tokens. We conduct preliminary experiments on the 3D molecule to text translation task using the PubChem dataset (Kim et al., 2019). Specifically, we compare our original 3D-MolT5, which sums 1D and 3D embeddings, with the sequential approach, where E3FP tokens are directly concatenated with 1D molecular tokens in a unified sequence. The results, presented in Table 9, show that the two approaches achieve comparable performance. However, the sequential approach significantly

---

[3] https://community.openai.com/t/why-9-11-is-larger-than-9-9-incredible/869824/5

Table 9: Results comparison between sum of 1D and 3D embedding versus sequential concatenation of 1D and 3D tokens.

| SETTING | BLEU-2 | BLEU-4 | ROUGE-1 | ROUGE-2 | ROUGE-L | METEOR |
|---|---|---|---|---|---|---|
| Embedding Summation | 37.78 | 29.61 | 42.94 | 27.51 | 37.40 | 38.86 |
| Sequential Concatenation | 37.65 | 29.34 | 43.25 | 27.64 | 37.45 | 38.98 |

increases the input sequence length, particularly for larger molecules, leading to higher computational costs and longer training times. In our experiments, the sequential concatenation requires more than 1.5 times the training time to converge compared to the embedding summation. Despite these drawbacks, the sequential approach offers practical advantages as it does not require modifications to the model's source code and aligns more naturally with tasks involving 3D molecular generation.

This finding highlights the flexibility of the E3FP-based framework and suggests potential extensions for tasks that benefit from sequential integration, such as 3D structure generation. Future work will explore the incorporation of these methods to further expand the capabilities of 3D-MolT5.

## D  MODEL CONFIGURATION

3D-MolT5 adopts the same architecture as T5 model (Raffel et al., 2020) with T5-1.1-base[4] configuration. The encoder and decoder have 12 layers. The dimensions of attention and feed-forward layers are 768 and 2048, respectively. The number of attention heads is 12. The size of the 1D vocabulary is 35,045, including original text tokens of T5 and additional SELFIES tokens, and the size of the 3D vocabulary is 4096. The total number of parameters of 3D-MolT5 is 255M. We use *nanoT5*[5] (Nawrot, 2023) as our codebase.

## E  PRE-TRAINING

### E.1  TRAINING TASK

As introduced in Section 3.4, the pre-training includes the denoising and translation tasks. We give the corresponding loss functions as follows.

**Denoising Tasks.** Given a sequence $X = \{x_i\}_{i=0}^{n-1}$, some consecutive spans of $X$ are randomly masked by sentinel tokens, and the model learns to reconstruct these spans:

$$\mathcal{L}_{\mathrm{D}} = - \sum_{t=0}^{|M|-1} \log P(X_M \mid X_{\setminus M}), \qquad (1)$$

where $X_M$ are the tokens that need to be recovered/generated, $|M|$ is the number of masked tokens, and $X_{\setminus M}$ is the input $X$ with the masked spans replaced by sentinel tokens.

**Translation Tasks.** In addition to denoising tasks, we also add translation tasks between modalities to enhance the representation learning,

$$\mathcal{L}_{\mathrm{T}} = - \sum_{t=0}^{|Y|-1} \log P(Y \mid X), \qquad (2)$$

where $X$ is the input sequence, such as 3D molecule tokens, and $Y$ is the target output sequence, such as the 1D text sequence.

### E.2  DATA AND CONFIGURATION

The pre-training is done on eight NVIDIA 80GB A100 GPUs. The total number of steps for pre-training is 400,000, with warm-up steps set to 10,000. AdamW (Loshchilov & Hutter, 2019) with

---

[4]https://huggingface.co/google/t5-v1_1-base
[5]https://github.com/PiotrNawrot/nanoT5

Root Mean Square (RMC) scaling optimizer is used. The peak learning rate is 2e-3 with cosine decay, and the minimum learning rate is 1e-5. The maximum length for input and output is 512. The batch size is set to 768. As shown in Table 10, the sizes of pre-training datasets vary significantly. To balance the data from different tasks during pre-training, we implement a *batch-level balancing strategy*. Each batch evenly includes data from all tasks, ensuring a more balanced and comprehensive pre-training process. For smaller datasets, such as molecule-text pairs from PubChem, we employ a round-robin strategy to repeat their usage multiple times, compensating for their limited size. For all molecular data, we first get its canonical SMILES from the provided SMILES or 3D structure using RDKit (Landrum et al., 2023), and then convert it to SELFIES using selfies toolkit (Krenn et al., 2020).[6] The resulting SELFIES are also wrapped by special tokens $\langle bom \rangle$ and $\langle eom \rangle$ to differentiate from text.

Table 10: Statistics of pre-training datasets. Deno. refers to T5 (Raffel et al., 2020) denoising task; Tran. refers to translation task. For the PubChem dataset, the 3D structure is obtained using the MMFF algorithm in RDKit (Landrum et al., 2023) and the text enriched by GPT-3.5 (OpenAI, 2023).

| DATA | TEXT | MOLECULE | | TASK | SIZE |
| | | 1D | 3D | | |
| --- | --- | --- | --- | --- | --- |
| PubChem SELFIES | - | ✓ | - | Deno. | 38,400,000 |
| C4-English | ✓ | - | - | Deno. | 2,210,000 |
| PubMed Central full articles | ✓ | - | - | Deno. | 38,400,000 |
| PubMed abstracts | ✓ | ✓ | - | Deno. | 33,404,528 |
| PCQM4Mv2 | - | ✓ | ✓ | Deno. & Tran. | 3,377,055 |
| PubChem molecule-text pairs | ✓ | ✓ | ✓ | Tran. | 298,861 |

# F  FINE-TUNING

Here we introduce more details about fine-tuning, including details about datasets and baselines. The fine-tuning is done on a single NVIDIA 80GB A100 GPU.

Details for datasets for fine-tuning are shown in Table 11. For all downstream datasets, we follow the same pipeline as described in Appendix E to first get the canonical SMILES from the provided SMILES or 3D structure using RDKit (Landrum et al., 2023), and then convert it to SELFIES wrapped by $\langle bom \rangle$ and $\langle eom \rangle$. **All reported results for 3D-MolT5 are the mean value obtained from three independent random runs.**

Table 11: Dataset statistics for donwstream fine-tuning. All the datasets are in instruction format. Small differences exist between our processed datasets and the original version, as we discard the data that can not be processed by E3FP (Axen et al., 2017).

| DATASET | MOLECULE | TASK | SIZE (TRAIN/VALIDATION/TEXT) |
| --- | --- | --- | --- |
| PubChemQC | 3D | Computed Property Prediction | 2,463,404/308,024/308,248 |
| QM9 | 3D | Computed Property Prediction | 347,774/1,928/1,928 |
| PubChem | 3D | Computed Property Prediction | 46,532/3,885/7,746 |
| | 3D | Descriptive Property Prediction | 59,775/4,980/9,940 |
| | 3D | 3D Molecule Captioning | 11,955/996/1988 |
| CheBI-20 | 1D | Text-based Molecule Generation | 26,407/3,301/3,300 |

**For PubChemQC (Maho, 2015) and PubChem (Kim et al., 2019) datasets**, we use the instruction versions built by 3D-MoLM (Li et al., 2023c). The *Generalist* version of 3D-MolT5 here is trained simultaneously on these two datasets with three types of tasks: computed property prediction, description property prediction, and 3D molecule captioning. We follow the same sampling algorithm as 3D-MoLM (Li et al., 2023c), where the sampling probabilities for each task are proportional to the

---

[6]https://github.com/aspuru-guzik-group/selfies

fourth root of the size of its data. For descriptive property prediction, the descriptive text is generated by employing GPT-3.5 (OpenAI, 2023) to read molecular captions and create five QA pairs for each molecule. The reported baseline results are derived from 3D-MoLM (Li et al., 2023c). Specifically, the baseline method 2D-MoLM is a variant of 3D-MoLM (Li et al., 2023c), where the 3D molecular encoder is replaced with a 2D molecular encoder. The baseline Llama2-7B (Touvron et al., 2023) directly removes the 3D molecular encoder of 3D-MoLM (Li et al., 2023c) and uses 1D SMILES as the molecular representation.

**For QM9 (Ruddigkeit et al., 2012) dataset**, we use its instruction version built by Mol-Instructions (Fang et al., 2023). The 3D structures are downloaded from DeepChem (Ramsundar et al., 2019). The *Generalist* version for QM9 is trained on the direct combination of its three subsets: HOMO, LUMO, and HOMO-LUMO gap. The reported baseline results are derived from BioT5+ (Pei et al., 2024a).

**For CheBI-20 (Edwards et al., 2022) dataset**, we manually convert it to instruction version. To avoid data leakage, we exclude the molecules of the CheBI-20 test set that are also present in the PubChem 3D molecule-text pairs in pre-training. In this task, molecular names are removed from the text to prevent the model from learning a simple mapping from molecular names to 1D sequences. The reported baseline results are mainly sourced from MolT5 (Edwards et al., 2022), MolReGPT (Li et al., 2023a), MolFM (Luo et al., 2023), GIT-Mol (Liu et al., 2024), MolXPT (Liu et al., 2023b), and BioT5 (Pei et al., 2023).

## G   CASE STUDY

The cases for computed molecular property prediction are shown in Table 12. We can find that 3D-MolT5 can give accurate numerical predictions about the computed properties of the input molecule. For descriptive property prediction, results in Table 13 show that 3D-MolT5 successfully answers the question about the composition of the input molecule, including the attached hexacosanoyl group and sphinganine backbone.

The case for 3D molecule captioning is shown in Table 14. 3D-MolT5 successfully predicts the molecular names, composition, pH, and functional relationship. The case for text-based molecule generation is shown in Table 15, where 3D-MolT5 generates the molecule that exactly matches the ground truth molecule.

## H   LIMITATIONS

In 3D-MolT5, the 3D structure information is only incorporated in the input, and 3D-MolT5 can not generate 3D molecular structure directly, which is mainly caused by two factors. (1) The hash algorithm and "folding" process are irreversible and may introduce value collisions, though the probability is small. (2) The pooling of embeddings at each iteration into the 3D embedding, and the pooling of 1D and 3D embeddings. The unified modeling of both the understanding and generation of 3D molecular structures remains an area for future exploration.

Table 12: Case studies for computed molecular property prediction task.

| MOLECULE | INSTRUCTION | 3D-MOLT5 | GROUND TRUTH |
|---|---|---|---|
| | I would like to know the Molecular Weight of this molecule, could you please provide it? If uncertain, provide an estimate. Respond with the numerical value only. | The Molecular Weight for the input molecule is 273.40 g/mol. | The Molecular Weight for the input molecule is 263.37 g/mol. |
| | I would like to know the LogP of this molecule, could you please provide it? If uncertain, provide an estimate. Respond with the numerical value only. | The LogP for the input molecule is 3.10. | The LogP for the input molecule is 2.80. |
| | I need to know the Topological Polar Surface Area of this molecule, could you please provide it? If uncertain, provide an estimate. Respond with the numerical value only. | The Topological Polar Surface Area for the input molecule is 49.30 $\text{Å}^2$. | The Topological Polar Surface Area for the input molecule is 49.30 $\text{Å}^2$. |
| | I need to know the Complexity of this molecule, could you please provide it? If uncertain, provide an estimate. Respond with the numerical value only. | The Complexity for the input molecule is 384.00. | The Complexity for the input molecule is 363.00. |
| | I need to know the HOMO of this molecule, could you please provide it? If uncertain, provide an estimate. Respond with the numerical value only. | The HOMO for the input molecule is -5.793 eV. | The HOMO for the input molecule is -5.769 eV. |
| | Please provide the LUMO value for this molecule. If uncertain, provide an estimate. Respond with the numerical value only. | The LUMO for the input molecule is 0.011 eV. | The LUMO for the input molecule is 0.054 eV. |
| | I am interested in the HOMO-LUMO Gap of this molecule, could you tell me what it is? If uncertain, provide an estimate. Respond with the numerical value only. | The HOMO-LUMO Gap for the input molecule is 5.810 eV. | The HOMO-LUMO Gap for the input molecule is 5.823 eV. |
| | Please provide the highest occupied molecular orbital (HOMO) energy of this molecule. | -0.2131. | -0.2132. |
| | What is the lowest unoccupied molecular orbital (LUMO) energy of this molecule? | -0.0066. | -0.0064. |
| | Can you give me the energy difference between the HOMO and LUMO orbitals of this molecule? | 0.2744. | 0.2756. |

Table 13: Case studies for descriptive molecular property prediction task.

| MOLECULE | INSTRUCTION | 3D-MOLT5 | GROUND TRUTH |
|---|---|---|---|
| | What is the composition of 1-O-(alpha-D-galactopyranosyl)-N-hexacosanoylsphinganine? | 1-O-(alpha-D-galactopyranosyl)-N-hexacosanoylsphinganine consists of a hexacosanoyl group attached to the nitrogen atom, with a galactosyl residue connected to the oxygen atom at position 1 of the sphinganine backbone. | 1-O-(alpha-D-galactopyranosyl)-N-hexacosanoylsphinganine is composed of a sphinganine backbone, a hexacosanoyl group attached to the nitrogen atom, and an alpha-D-galactopyranosyl residue at the O-1 position. |

Table 14: Case studies for 3D molecular captioning task.

| MOLECULE | INSTRUCTION | 3D-MOLT5 | GROUND TRUTH |
|---|---|---|---|
|  | Describe the input molecule. | 4-methylbenzoyl-AMP(1-) is an organophosphate oxoanion obtained by deprotonation of the phosphate OH group of 4-methylbenzoyl-AMP; major species at pH 7.3. It is functionally related to an adenosine 5'-monophosphate(2-) and a 4-methylbenzoate. | 4-methylbenzoyl-AMP(1-) is an organophosphate oxoanion obtained by deprotonation of the phosphate OH group of 4-methylbenzoyl-AMP; major microspecies at pH 7.3. It is functionally related to an adenosine 5'-monophosphate(2-) and a p-toluate. |

Table 15: Case studies for text-based molecular generation task.

| DESCRIPTION | INSTRUCTION | 3D-MOLT5 | GROUND TRUTH |
|---|---|---|---|
| The molecule is a member of the class of naphthoates that is 1-naphthoate substituted at positions 3 and 5 by hydroxy and methyl groups respectively; major species at pH 7.3. It has a role as a bacterial metabolite. It is a conjugate base of a 3-hydroxy-5-methyl-1-naphthoic acid. | Generate a molecule that fits the input description. | [C][C][=C][C][=C][Branch1][C][O-1][C][=C][Branch1][=Branch1][C][=Branch1][C][=O][O][C][Ring1][#Branch2][=C][C][=C][Ring1][=C]. | [C][C][=C][C][=C][Branch1][C][O-1][C][=C][Branch1][=Branch1][C][=Branch1][C][=O][O][C][Ring1][#Branch2][=C][C][=C][Ring1][=C]. |

