# OpenReview forum: "3D-MolT5: Leveraging Discrete Structural Information for Molecule-Text Modeling"
_ICLR.cc/2025/Conference — ICLR 2025 Poster_

### Official Review · Reviewer_mQNb · 2024-10-16

**Soundness:** 3
**Presentation:** 2
**Contribution:** 3
**Rating:** 6
**Confidence:** 4

**Summary:**

The authors propose 3D-MolT5, a unified framework designed to model molecule in both sequence and 3D structure spaces. The key innovation of our approach lies in mapping fine-grained 3D substructure representations into a specialized 3D token vocabulary. This methodology facilitates the integration of sequence and structure representations in a tokenized format, enabling 3D-MolT5 to encode molecular sequences, molecular structures, and text sequences within a unified architecture. Leveraging this tokenized input strategy, we build a foundation model that unifies the sequence and structure data formats. We then conduct joint pre-training with multi-task objectives to enhance the model’s comprehension of these diverse modalities within a shared representation space. Authors also provide source code for reproducibility.

**Strengths:**

The authors describe developments for LLMs to capture textual information in molecules, as well as graph representation models to capture 3D structure topology of molecules. Limitations of the existing work is also well described.

There are a variety of tasks evaluated both on text generation as well as structure learning, which is informative and comprehensive.

**Weaknesses:**

The paper seems to be missing the important citation from the IJCAI survey paper on Graph-based molecular representation learning:

[IJCAI 2023] Zhichun Guo, Kehan Guo, Bozhao Nan, Yijun Tian, Roshni G. Iyer, Yihong Ma, Olaf Wiest, Xiangliang Zhang, Wei Wang, Chuxu Zhang, and Nitesh V. Chawla. 2023. Graph-based molecular representation learning. In Proceedings of the Thirty-Second International Joint Conference on Artificial Intelligence (IJCAI '23). Article 744, 6638–6646. https://doi.org/10.24963/ijcai.2023/744

While experiment results achieve competitive results, there seem to be only 2 benchmark datasets being considered. There are more diverse and difficult datasets such as USPTO. Can the authors evaluate on more datasets to make the results more conclusive?

**Questions:**

See above section for details.

---

> ### Author Response · Authors · 2024-11-19
>
> Thanks for your review. We provide the following responses to address your concerns.
> > W1: The paper seems to be missing the important citation from the IJCAI survey paper on Graph-based molecular representation learning [1]
> >
>
> Thanks for bringing this important paper to our attention. We recognize the relevance of this survey[1] on graph-based molecular representation learning and its contribution to this field. We have appropriately cite this work in the related work section of the revised version of our manuscript.
>
> > W2: While experiment results achieve competitive results, there seem to be only 2 benchmark datasets being considered. There are more diverse and difficult datasets such as USPTO. Can the authors evaluate on more datasets to make the results more conclusive?
> >
>
> Thank you for this valuable suggestion. Following your recommendation, we extended our evaluation to include additional benchmark datasets, such as **USPTO-50k[2]**, **Mol-Instructions[3] (including 3 tasks: forward reaction prediction, reagent prediction, and retrosynthesis)**, and **MoleculeNet[4]**, to further validate the effectiveness of our approach. The results of these experiments are shown in the following Table 1-5.
>
> Table 1: Results for the retrosynthesis task on USPTO-50k[2].
>
> |  | BLEU-2 ↑ | Levenshtein ↓ | RDKit FTS ↑ | Validity ↑ |
> | --- | --- | --- | --- | --- |
> | ReactionT5 | 81.63 | 17.69 | 0.7400 | 97.58 |
> | LlaSMol-7B | 50.09 | 31.28 | 0.7351 | 99.65 |
> | 3D-MoLM | 81.31 | 16.21 | 0.7341 | 90.31 |
> | MolX | 82.59 | 15.74 | 0.7466 | 92.19 |
> | 3D-MolT5 | **86.23** | **14.08** | **0.7538** | **100.00** |
>
> Table 2: Results for the forward reaction prediction task on Mol-Instructions[3].
>
> |  | Exact Match ↑ | BLEU ↑ | Levenshtein ↓ | RDK FTS ↑ | MACCS FTS ↑ | Morgan FTS ↑ | Validity ↑ |
> | --- | --- | --- | --- | --- | --- | --- | --- |
> | Mol-Instructions | 0.045 | 0.654 | 27.262 | 0.313 | 0.509 | 0.262 | 1.000 |
> | InstructMol | 0.536 | 0.967 | 10.851 | 0.776 | 0.878 | 0.741 | 1.000 |
> | UniMoT | 0.611 | 0.980 | 8.297 | 0.836 | 0.911 | 0.807 | 1.000 |
> | 3D-MolT5 | **0.879** | **0.994** | **2.842** | **0.955** | **0.978** | **0.942** | 1.000 |
>
> Table 3: Results for the reagent prediction task on Mol-Instructions[3].
>
> |  | Exact Match ↑ | BLEU ↑ | Levenshtein ↓ | RDK FTS ↑ | MACCS FTS ↑ | Morgan FTS ↑ | Validity ↑ |
> | --- | --- | --- | --- | --- | --- | --- | --- |
> | Mol-Instructions | 0.044 | 0.224 | 23.167 | 0.237 | 0.364 | 0.213 | 1.000 |
> | InstructMol | 0.129 | 0.610 | 19.644 | 0.444 | 0.539 | 0.400 | 1.000 |
> | UniMoT | 0.167 | 0.728 | 14.588 | 0.549 | 0.621 | 0.507 | 1.000 |
> | 3D-MolT5 | **0.277** | **0.756** | **13.173** | **0.571** | **0.646** | **0.540** | 1.000 |
>
> Table 4: Results for the retrosynthesis task on Mol-Instructions[3].
>
> |  | Exact Match ↑ | BLEU ↑ | Levenshtein ↓ | RDK FTS ↑ | MACCS FTS ↑ | Morgan FTS ↑ | Validity ↑ |
> | --- | --- | --- | --- | --- | --- | --- | --- |
> | Mol-Instructions | 0.009 | 0.705 | 31.227 | 0.283 | 0.487 | 0.230 | 1.000 |
> | InstructMol | 0.407 | 0.941 | 13.967 | 0.753 | 0.852 | 0.714 | 1.000 |
> | UniMoT | 0.478 | 0.974 | 11.634 | 0.810 | 0.909 | 0.771 | 1.000 |
> | 3D-MolT5 | **0.679** | **0.974** | **6.201** | **0.907** | **0.940** | **0.878** | 1.000 |
>
> Table 5: Results (AUROC) for the molecular property prediction task on MoleculeNet[4].
>
> |  | BACE | BBBP | HIV | ClinTox | Avg |
> | --- | --- | --- | --- | --- | --- |
> | GraphCL | 75.4 | 69.7 | 78.5 | 76.0 | 74.9 |
> | GraphMVP-C | 81.2 | 72.4 | 77.0 | 77.5 | 77.0 |
> | MGSSL | 79.7 | 70.5 | 79.5 | 80.7 | 77.6 |
> | MolCLR | **89.0** | 73.8 | 80.6 | 93.2 | 84.2 |
> | GEM | 85.6 | 72.4 | 80.6 | 90.1 | 82.2 |
> | Uni-Mol | 85.7 | 72.9 | **80.8** | 91.9 | 82.8 |
> | KV-PLM | 71.9 | 66.9 | 68.8 | 84.3 | 73.0 |
> | MoMu | 76.7 | 70.5 | 75.9 | 79.9 | 75.8 |
> | MolFM | 83.9 | 72.9 | 78.8 | 79.7 | 78.8 |
> | Galactica-6.7B | 58.4 | 53.5 | 72.2 | 78.4 | 65.6 |
> | Galactica-30B | 72.7 | 59.6 | 75.9 | 82.2 | 72.6 |
> | Galactica-120B | 61.7 | 66.1 | 74.5 | 82.6 | 71.2 |
> | UniMoT | 83.7 | 71.4 | 78.5 | 92.9 | 81.6 |
> | 3D-MolT5 | 88.1 | **76.5** | **80.8** | **95.4** | **85.2** |
>
> As shown in the results, our method achieves strong and consistent performance across these datasets, demonstrating its robustness and versatility in handling diverse and challenging molecular tasks. The complete version of results above have been included in **Appendix B and Table 6-8** (**Page 19-21**) in the revised version of our manuscript.

---

> > ### Author Response · Authors · 2024-11-19
> >
> > **References**
> >
> > [1] Guo, Zhichun, et al. "Graph-based molecular representation learning." *Proceedings of the Thirty-Second International Joint Conference on Artificial Intelligence*. 2023.
> >
> > [2] Schneider, Nadine, Nikolaus Stiefl, and Gregory A. Landrum. "What’s what: The (nearly) definitive guide to reaction role assignment." *Journal of chemical information and modeling* 56.12 (2016): 2336-2346.
> >
> > [3] Fang, Yin, et al. "Mol-Instructions: A Large-Scale Biomolecular Instruction Dataset for Large Language Models." *The Twelfth International Conference on Learning Representations*.
> >
> > [4] Wu, Zhenqin, et al. "MoleculeNet: a benchmark for molecular machine learning." *Chemical science* 9.2 (2018): 513-530.

---

> > > ### Comment · Reviewer_mQNb · 2024-11-27
> > > **Response to APC**
> > >
> > > Thank you for your insightful feedback. Please find a detailed response to your questions below.
> > >
> > > Q1: Your suggestion to include this relevant citation is valuable. To make your comment more actionable, consider explaining why you think this survey paper is particularly important for this work. How does it relate to or potentially impact the contributions of the current paper? You could also ask the authors to discuss how their approach compares to or differs from the graph-based approaches covered in this survey.
> > >
> > > A1: The ultimate goal for works in this space is to design an improved and more effective molecular representation model whether it be via text only or a combination of text and graph structure. This paper is highly relevant to this research work since it is a comprehensive survey of recent works in the space of molecular modeling both through text and graph based approaches. This paper also highlights the pitfalls of several LLM only based models for molecular representation and the reason behind why SMILE strings and 2D/3D representations are also required.
> > >
> > > Q2: The USPTO dataset is more diverse and difficult because of its sheer size and the variety of molecular reactions it captures. This dataset also contains rich, multi-modal information compared to other datasets as it includes textual information and details about the experiment condition like temperature, enzymes and catalysts involved, quantity etc., as well as graph structure information via SMILE string data. It is also a difficult dataset because the best known performance on these models is around 60%, compared to other benchmarks where the performance is 90+%. As such, there is much more room for improvement on such datasets, and will better distinguish how effective model representation performance is.

---

> > > > ### Author Response · Authors · 2024-11-28
> > > >
> > > > > The ultimate goal for works in this space is to design an improved and more effective molecular representation model whether it be via text only or a combination of text and graph structure. This paper is highly relevant to this research work since it is a comprehensive survey of recent works in the space of molecular modeling both through text and graph based approaches. This paper also highlights the pitfalls of several LLM only based models for molecular representation and the reason behind why SMILE strings and 2D/3D representations are also required.
> > > > >
> > > >
> > > > We appreciate your recognition of the relevance of our work to the broader research in molecular representation learning. However, we would like to clarify that the survey paper you mentioned primarily focuses on **graph-based molecular representation learning**, covering 2D, 3D molecular graphs and knowledge graphs, as well as the corresponding representation learning models. While it provides valuable insights into these graph-based approaches, it does not appear to include any discussion or exploration of **molecule-text joint modeling**, which is the primary focus of our paper.
> > > >
> > > > Besides, we are confused of the rejection reason by missing a related survey paper while our work is a method work. We will cover this survey paper as mentioned before and we hope reviewer could reconsider our contribution.
> > > >
> > > > > The USPTO dataset is more diverse and difficult because of its sheer size and the variety of molecular reactions it captures. This dataset also contains rich, multi-modal information compared to other datasets as it includes textual information and details about the experiment condition like temperature, enzymes and catalysts involved, quantity etc., as well as graph structure information via SMILE string data. It is also a difficult dataset because the best known performance on these models is around 60%, compared to other benchmarks where the performance is 90+%. As such, there is much more room for improvement on such datasets, and will better distinguish how effective model representation performance is.
> > > > >
> > > >
> > > > Thanks for highlighting the significance of the USPTO dataset. We agree that and we have done experiments in previous rebuttal, our 3D-MolT5 **achieves strong results on this challenging dataset**, as shown in **Table 1** (USPTO-50k) and **Table 2-4** (Mol-Instructions, which is also sourced from the USPTO database) in [previous rebuttal message](https://openreview.net/forum?id=eGqQyTAbXC&noteId=YB4wF1X2a9). We believe these **results provide solid evidence of the effectiveness of our approach on the USPTO dataset**, particularly in handling the diverse and multi-modal nature of the data.

---

> ### Comment · Reviewer_mQNb · 2024-11-27
> **Response to Authors**
>
> Thanks for your detailed response. Upon reviewing all of the feedback and other reviews, I have decided to keep my score unchanged. While experiment results do seem promising, I feel that the model is lacking in novelty and its approaches have been explored within the research community.

---

> > ### Author Response · Authors · 2024-11-28
> >
> > > While experiment results do seem promising, I feel that the model is lacking in novelty and its approaches have been explored within the research community.
> > >
> >
> > Our work introduces several key innovations that highly distinguish it from prior approaches.
> >
> > - We propose **discrete tokenization of 3D molecular structures by using E3FP algorithm**, a novel and critical contribution that allows 3D molecular data to be seamlessly integrated into the same framework as 1D molecular sequences and text. Unlike previous models that rely on continuous 3D data. This tokenization allows for **early and extensive cross-modal interaction**, which is crucial for tasks requiring joint understanding of 1D, 3D molecular, and textual modalities.
> > - Our model does not require **external encoders** **and multi-stage pre-training**, as seen in many prior approaches, but instead integrates 1D and 3D modalities directly within the same pre-training framework. This unified approach simplifies the architecture and enhances the efficiency of the training process, eliminating the need for alignment between separate models.
> > - Our experiments demonstrate that **3D-MolT5** outperforms existing methods in tasks such as **3D molecular property prediction and captioning**, as well as **chemical reaction tasks as you mentioned**, providing further evidence of the effectiveness and novelty of our approach.
> > - **Notably, we are almost the first one to model 3D molecular structures with discrete tokens. If you could provide similar papers that are in same novelty as ours, we are happy to provide discussions.**
> >
> > We believe that these contributions represent a significant step forward in molecular modeling and make a valuable addition to the research community.

---

> ### Comment · Reviewer_mQNb · 2024-11-29
>
> Thanks for highlighting the contributions of your work and its distinction to prior works. This has addressed my concerns and I have raised my score to a 6. If you read my above review accordingly my concern was primarily based on novelty not on a minor feedback of including some more recent 2023 citation work since that makes no sense.
>
> Side Note for feedback to authors: I share Reviewer NdAP's concern regarding the choice of an aggressive tone in the rebuttal. It is in everyone's best interest for feedback to be approached constructively, rather than perceived as a "personal attack." Only by fostering a collaborative and respectful environment can our research community continue to progress and produce high-quality work. I would like to express that if this behavior continues, I may feel compelled to flag it for review by the APC. It is important that we maintain a respectful and constructive tone in all interactions to foster a productive and inclusive environment.

---

> > ### Author Response · Authors · 2024-11-29
> >
> > Thanks for your valuable feedback and for raising the score. We are delighted that our responses has addressed your concerns, especially regarding the novelty of our work. We sincerely appreciate your recognition. We would also like to apologize if the tone of our rebuttal came across as aggressive or inappropriate. That was never our intention, and we deeply respect the importance of fostering a collaborative and respectful dialogue within the research community. We will certainly take your feedback and ensure that our future communications remain constructive.

---

### Official Review · Reviewer_NdAP · 2024-10-30

**Soundness:** 3
**Presentation:** 3
**Contribution:** 2
**Rating:** 6
**Confidence:** 4

**Summary:**

The paper introduces a unique framework, 3D-MolT5, that integrates 3D structural data and molecular sequences into a unified language model. 3D-MolT5 embeds molecular geometry directly into the language model using discrete 3D tokens generated from the E3FP algorithm. The model is pre-trained with multi-task objectives, including both denoising and translation tasks. The model achieves state-of-the-art results in tasks like molecular property prediction, molecule captioning, and text-based molecule generation.

**Strengths:**

- The introduction of 3D tokens for representing molecular substructures at atomic level removes the dependency on structure encoders.
- The combination of 1D and 3D embeddings enables alignment of molecular sequence and structural data at atomic level.
- The novel pre-training denoising and translation tasks enhance cross-modal training and alignment.
- The model demonstrates performance gains over existing baselines across different benchmarks, including 3D-dependent tasks.

**Weaknesses:**

- The term “tokenization” may not be accurate here, as 3D-MolT5 does not achieve unified molecule-text modeling in token space as UniMoT [1] does. UniMoT can generate molecule tokens as text, but 3D-MolT5 lacks the ability to generate 3D tokens. The token generation ability is the major difference between discrete and continuous structures, with 3D-MolT5 limited to encoding rather than generating 3D data.

  - The 3D “tokenization” in this paper functions more like a 3D molecular encoder, encoding each atom into a 3D embedding rather than generating 3D structures.
  - 3D-MolT5 lacks 3D generation abilities because it cannot map 3D embeddings back to specific atoms or structures.
  - The motivation section resembles that of UniMoT. A more detailed comparison between the two would be helpful.

- The E3FP algorithm introduces additional space and time complexity than using a well-pretrained encoder.

- The reasoning for further encoding $d_i$ with $E_{3D}(d_i)$ instead of using $d_i$ directly is unclear.

- It’s not explained why the authors chose the E3FP algorithm over a 3D molecular encoder like Uni-Mol. And the paper does not provide an ablation comparison to illustrate any advantages of E3FP.

- In Algorithm 1, it appears that $\cup$ is used to indicate concatenation; this should be stated for clarity.

[1] Juzheng Zhang, Yatao Bian, Yongqiang Chen, and Quanming Yao. Unimot: Unified molecule-text language model with discrete token representation. arXiv preprint arXiv:2408.0086.

**Questions:**

- Why do the authors focus solely on T5 instead of exploring other LLM architectures? Could the 3D "tokenization" approach work effectively with models like Llama or Mistral?

- What advantages does the E3FP algorithm have over a 3D molecular encoder?

- How many embeddings are used for encoding SELFIES tokens and 3D tokens?

- Since the SELFIES sequence length differs from the number of atoms, how do you manage the additional SELFIES tokens?

- What are the time and memory costs for running the E3FP algorithm?

---

> ### Author Response · Authors · 2024-11-19
>
> Thank you for your insightful comments regarding our paper. We appreciate your time and effort and would like to address the concerns with following responses.
> > W1: The term “tokenization” may not be accurate here, as 3D-MolT5 does not achieve unified molecule-text modeling in token space as UniMoT [1] does. UniMoT can generate molecule tokens as text, but 3D-MolT5 lacks the ability to generate 3D tokens. The token generation ability is the major difference between discrete and continuous structures, with 3D-MolT5 limited to encoding rather than generating 3D data.
> >
> > - The 3D “tokenization” in this paper functions more like a 3D molecular encoder, encoding each atom into a 3D embedding rather than generating 3D structures.
> > - 3D-MolT5 lacks 3D generation abilities because it cannot map 3D embeddings back to specific atoms or structures.
> > - The motivation section resembles that of UniMoT. A more detailed comparison between the two would be helpful.
>
> Thanks for insightful comments. Your primary focus seems to be on the definition of "3D tokenization" and its relationship with generation, as well as the comparison between our approach and UniMoT. We provide following points in detail.
>
> - We use the term **"tokenization"** to describe the process of converting molecule or text into smaller units, or tokens. For example, text→subwords, 1D molecule sequence→SELFIES tokens,  and our 3D molecular structures→3D tokens. If you strongly insist that "tokenization" should imply the ability to perform generation (i.e., a reversible detokenization process), we are open to revisiting this terminology.
> - Currently, our design focuses primarily on **molecular understanding tasks**, and we acknowledge that 3D-MolT5 cannot generate 3D molecular structures. Future extensions may enable the model to generate E3FP tokens, which could then be input into a molecule generation model to produce 3D structures.
> - For the relevance to UniMoT[1], the differences are as follows:
>     - **Modalities modeled.** Our work focuses on incorporating 3D molecular structure modeling, which is absent in UniMoT. Specifically, 3D-MolT5 accepts 1D molecules, 3D molecules, and text as inputs, and can output 1D molecules or text. In contrast, UniMoT models 1D molecules, 2D molecular graphs, and text, without handling 3D molecular structures.
>     - **Generative capabilities.** Both models share similar generative capabilities for 1D molecular sequences and text. While 3D-MolT5 does not currently support 3D structure generation, this does not represent a limitation relative to UniMoT, which also lacks 3D generation abilities.
>     - **Training workflow.** The training process for 3D-MolT5 is simpler, consisting of just two stages: multi-task pre-training followed by fine-tuning. UniMoT, by contrast, involves a more complex four-stage process: Causal Q-Former pre-training, molecule tokenizer pre-training, unified molecule-text pre-training, and task-specific instruction tuning.
>     - **Task coverage and performance.** Both models are capable of performing tasks related to 1D and 2D molecules, and we have included additional benchmarks in the revised manuscript for a direct comparison. Results shown in following Table 1-5 demonstrate superior overall performance compared to UniMoT on overlapping tasks. Furthermore, 3D-MolT5 uniquely addresses 3D-related tasks, such as 3D molecular property prediction, which UniMoT cannot perform.
>
> We hope this clarifies the key differences and strengths of 3D-MolT5 relative to UniMoT. The complete version of the results have been elaborated in **Appendix B** and **Table 6-8** (**Page 19-21**) in the revised manuscript, and we appreciate your feedback in helping us refine our paper.
>
> Table 1: Results for the forward reaction prediction task on Mol-Instructions[2].
>
> |  | Exact Match ↑ | BLEU ↑ | Levenshtein ↓ | RDK FTS ↑ | MACCS FTS ↑ | Morgan FTS ↑ | Validity ↑ |
> | --- | --- | --- | --- | --- | --- | --- | --- |
> | Mol-Instructions | 0.045 | 0.654 | 27.262 | 0.313 | 0.509 | 0.262 | 1.000 |
> | InstructMol | 0.536 | 0.967 | 10.851 | 0.776 | 0.878 | 0.741 | 1.000 |
> | UniMoT | 0.611 | 0.980 | 8.297 | 0.836 | 0.911 | 0.807 | 1.000 |
> | 3D-MolT5 | **0.879** | **0.994** | **2.842** | **0.955** | **0.978** | **0.942** | 1.000 |
>
> Table 2: Results for the reagent prediction task on Mol-Instructions[2].
>
> |  | Exact Match ↑ | BLEU ↑ | Levenshtein ↓ | RDK FTS ↑ | MACCS FTS ↑ | Morgan FTS ↑ | Validity ↑ |
> | --- | --- | --- | --- | --- | --- | --- | --- |
> | Mol-Instructions | 0.044 | 0.224 | 23.167 | 0.237 | 0.364 | 0.213 | 1.000 |
> | InstructMol | 0.129 | 0.610 | 19.644 | 0.444 | 0.539 | 0.400 | 1.000 |
> | UniMoT | 0.167 | 0.728 | 14.588 | 0.549 | 0.621 | 0.507 | 1.000 |
> | 3D-MolT5 | **0.277** | **0.756** | **13.173** | **0.571** | **0.646** | **0.540** | 1.000 |

---

> > ### Author Response · Authors · 2024-11-19
> >
> > Table 3: Results for the retrosynthesis task on Mol-Instructions[2].
> >
> > ||Exact Match ↑|BLEU ↑|Levenshtein ↓|RDK FTS ↑|MACCS FTS ↑|Morgan FTS ↑|Validity ↑|
> > |---|---|---|---|---|---|---|---|
> > |Mol-Instructions|0.009|0.705|31.227|0.283|0.487|0.230|1.000|
> > |InstructMol|0.407|0.941|13.967|0.753|0.852|0.714|1.000|
> > |UniMoT|0.478|0.974|11.634|0.810|0.909|0.771|1.000|
> > |3D-MolT5|**0.679**|**0.974**|**6.201**|**0.907**|**0.940**|**0.878**|1.000|
> >
> > Table 4: Results for the 3D molecule captioning task on PubChem[3].
> >
> > || BLEU-2 ↑ | BLEU-4 ↑ | ROUGE-1 ↑ | ROUGE-2 ↑ | ROUGE-L ↑ | METEOR ↑ |
> > |---|---|---|---|---| --- | --- |
> > | MolT5-Large | 25.9 | 17.3 | 34.1 | 16.4 | 23.4 | 28.0 |
> > | MoMu-Large | 26.3 | 18.0 | 34.8 | 16.9 | 24.8 | 28.7 |
> > | 3D-MoLM | 30.3 | 22.5 | 36.8 | 22.3 | 31.2 | 33.1 |
> > | MolX | 31.4 | 24.3 | 44.2 | 29.0 | 38.8 | 39.6 |
> > | UniMoT | 31.3 | 23.8 | 37.5 | 23.7 | 33.6 | 34.8 |
> > | 3D-MolT5 | **42.1** | **34.2** | **48.1** | **33.2** | **42.3** | **44.7** |
> >
> > Table 5: Results (AUROC) for the molecular property prediction task on MoleculeNet[4].
> >
> > |  | BACE | BBBP | HIV | ClinTox | Avg |
> > | --- | --- | --- | --- | --- | --- |
> > | MolCLR | **89.0** | 73.8 | 80.6 | 93.2 | 84.2 |
> > | Uni-Mol | 85.7 | 72.9 | **80.8** | 91.9 | 82.8 |
> > | MolFM | 83.9 | 72.9 | 78.8 | 79.7 | 78.8 |
> > | Galactica | 72.7 | 59.6 | 75.9 | 82.2 | 72.6 |
> > | UniMoT | 83.7 | 71.4 | 78.5 | 92.9 | 81.6 |
> > | 3D-MolT5 | 88.1 | **76.5** | **80.8** | **95.4** | **85.2** |
> >
> > > W2: The E3FP algorithm introduces additional space and time complexity than using a well-pretrained encoder.
> >
> > Thanks for raising this point. It is important to clarify that the process of converting a molecule's 3D structure into E3FP tokens is performed **offline** before model training or inference. This preprocessing step ensures that the tokenization of 3D structures does not introduce additional space or time complexity during model training or inference.
> >
> > > W3: The reasoning for further encoding $\bf d_i$ with $E_{3D}(\bf d_i)$ instead of using $\bf d_i$ directly is unclear.
> > >
> >
> > It seems there may be a misunderstanding regarding the notation. In our framework, $\bf d_i$ represents the 3D token, which is essentially an embedding index. The notation $E_{3D}(\bf d_i)$ refers to the process of indexing into the 3D embedding layer using the 3D token to obtain the corresponding 3D embedding. This 3D embedding is then combined with the 1D embedding to produce the final embedding, which serves as the input to the T5 model.
> >
> > > W4: It’s not explained why the authors chose the E3FP algorithm over a 3D molecular encoder like Uni-Mol. And the paper does not provide an ablation comparison to illustrate any advantages of E3FP.
> > >
> > - **The reason of choose E3FP algorithm over a 3D molecular encoder**
> >     - Our choice to use the E3FP algorithm is primarily motivated by its ability to **transform a molecule's 3D structure into discrete tokens**, where each E3FP token corresponds to a specific substructure. This discrete tokenization allows us to model 1D molecular sequences, 3D molecular structures, and text within a unified architecture, facilitating seamless integration of these modalities.
> >     - In contrast, Uni-Mol[5] encodes molecules as atom-wise embeddings or representations, where the same atom can have different representations in different positions or molecules. This results in an **unbounded representation space**, making it **infeasible to encode 3D structures into a finite token space, as E3FP does**. **Such a finite token space is crucial for enabling a unified modeling framework.**
> > - **Ablation comparison**
> >     - The superior performance of 3D-MolT5 over 3D-MoLM provides evidence of the advantages of E3FP-based discrete tokenization. In 3D-MoLM[3], Uni-Mol is used as the external 3D encoder, Llama-2[6] as the language decoder, and they are connected through Q-Former. However, this approach of leveraging separate models for different modalities has inherent limitations:
> >         - **Requiring additional components and processes**, such as alignment mechanisms or projection layers, to ensure compatibility between the representations of different modalities. Such late-stage alignment processes are challenging to optimize effectively.
> >         - The alignment is further difficult particularly when the **paired data between text and 3D structures are limited**.
> >         - Treating modalities separately **hinders early and rich cross-modal interactions**, which are critical for enhancing joint representations and ensuring a seamless fusion of information.
> >     - In contrast, 3D-MolT5 leverages E3FP to encode 3D information into discrete tokens compatible with the pre-training framework of language models. This eliminates the need for external encoders and late-stage alignment, **simplifying the architecture while enhancing cross-modal interactions during pre-training**. As a result, 3D-MolT5 demonstrates better performance across downstream tasks, highlighting the benefits of our unified approach.

---

> ### Author Response · Authors · 2024-11-19
>
> > W5: In Algorithm 1, it appears that ∪ is used to indicate concatenation; this should be stated for clarity.
> >
>
> Thanks for pointing this out. We appreciate your careful review. In the revised manuscript, we have explicitly stated that **∪** in **Algorithm 1** (**Page 5**) is used to indicate concatenation to ensure clarity and avoid any potential confusion.
>
> > Q1: Why do the authors focus solely on T5 instead of exploring other LLM architectures? Could the 3D "tokenization" approach work effectively with models like Llama or Mistral?
> >
>
> Thanks for raising this question. We chose to focus on T5 as the base architecture due to its established success in molecule-text modeling (e.g., MolT5[7], BioT5[8], nach0[9]) . T5 leverages bidirectional attention mechanism for effective cross-modal understanding.
>
> The proposed E3FP-based 3D tokenization approach is designed to be model-agnostic ****and can be applied to other language models, including GPT-based architectures like Llama or Mistral. However, due to time constraints, we focused on large-scale pre-training from scratch using T5 in this work. We will explore the effectiveness of our tokenization approach with GPT-based models in the future.
>
> > Q2: What advantages does the E3FP algorithm have over a 3D molecular encoder?
> >
>
> **Similar to previous answer in W4**, the primary advantage of using the E3FP algorithm over a 3D molecular encoder lies in its alignment with our motivation to develop a **discrete and unified model** that does not require external alignment between different modalities. This approach ensures a more seamless and effective integration of 1D molecular sequences, 3D molecular structures, and text.
>
> > Q3: How many embeddings are used for encoding SELFIES tokens and 3D tokens?
> >
>
> In 3D-MolT5, we use **2944 embeddings for encoding SELFIES tokens** and **4097 embeddings for encoding 3D tokens**. We have included these details in **Appendix A.4** (**Page 17-18**) in the revised manuscript for clarity and completeness.
>
> > Q4: Since the SELFIES sequence length differs from the number of atoms, how do you manage the additional SELFIES tokens?
> >
>
> Thank you for this insightful question. As you correctly pointed out, the SELFIES sequence length differs from the number of atoms due to the presence of **structure directive tokens** such as `[Ring1]` and `[=Branch1]`. In 3D-MolT5, these special SELFIES tokens are preserved in the 1D sequence and indexed into the 1D embedding layer as usual. Since these tokens do not correspond to actual atoms and therefore lack 3D structural information, their associated 3D token is assigned a value of `-1`, and the corresponding 3D embedding is a zero vector, as described in **Figure 2** and detailed in **Appendix A.4**. We have updated the revised manuscript in **Appendix A.4** (**Page 17-18**) to further clarify this process and ensure that the handling of these tokens is clearly described.
>
> > Q5: What are the time and memory costs for running the E3FP algorithm?
> >
>
> In practice, **the generation of E3FP fingerprints is performed offline using CPU** and can be executed rapidly through **multi-processing**, achieving a throughput of approximately **300 samples per second** using 24 parallel processes. The memory cost for processing each sample is **0.4 MB**.
>
> **References**
>
> [1] Zhang, Juzheng, et al. "Unimot: Unified molecule-text language model with discrete token representation." *arXiv preprint arXiv:2408.00863* (2024).
>
> [2] Fang, Yin, et al. "Mol-Instructions: A Large-Scale Biomolecular Instruction Dataset for Large Language Models." *The Twelfth International Conference on Learning Representations*.
>
> [3] Li, Sihang, et al. "Towards 3D Molecule-Text Interpretation in Language Models." *The Twelfth International Conference on Learning Representations*.
>
> [4] Wu, Zhenqin, et al. "MoleculeNet: a benchmark for molecular machine learning." *Chemical science* 9.2 (2018): 513-530.
>
> [5] Zhou, Gengmo, et al. "Uni-Mol: A Universal 3D Molecular Representation Learning Framework." *The Eleventh International Conference on Learning Representations*.
>
> [6] Touvron, Hugo, et al. "Llama 2: Open foundation and fine-tuned chat models." *arXiv preprint arXiv:2307.09288* (2023).
>
> [7] Edwards, Carl, et al. "Translation between Molecules and Natural Language." *Proceedings of the 2022 Conference on Empirical Methods in Natural Language Processing*. 2022.
>
> [8] Pei, Qizhi, et al. "BioT5: Enriching Cross-modal Integration in Biology with Chemical Knowledge and Natural Language Associations." *Proceedings of the 2023 Conference on Empirical Methods in Natural Language Processing*. 2023.
>
> [9] Livne, Micha, et al. "nach0: Multimodal natural and chemical languages foundation model." *Chemical Science* 15.22 (2024): 8380-8389.

---

> > ### Comment · Reviewer_NdAP · 2024-11-24
> >
> > Thank you very much for your detailed response. However, I feel that some of my concerns remain unaddressed.
> >
> > > a discrete and unified model
> >
> > The authors repeatedly claim in the rebuttal and manuscript that the proposed model is "unified." However, I do not agree with this characterization. In the context of multi-modal LLMs, "unified" typically refers to unifying comprehension and generation capabilities. Your model does not support the generation of 3D tokens, so it cannot truly be considered "unified." Instead, it would be more accurate to state that your model uses 3D tokens in a discrete form.
> >
> > > This results in an unbounded representation space, making it infeasible to encode 3D structures into a finite token space, as E3FP does. Such a finite token space is crucial for enabling a unified modeling framework.
> >
> > An unbounded representation space should not necessarily be viewed as a disadvantage. It allows for more diverse representations compared to a finite token space and often results in better performance than discrete architectures in comprehension tasks.
> >
> > The primary purpose of discretization is to enable generation, as demonstrated in various multi-modal LLM works. This often involves sacrificing some comprehension performance to achieve generative capabilities. The authors did not clearly justify the choice of the E3FP algorithm over a 3D molecular encoder.
> >
> > Additionally, it is difficult to claim that the discretization process can be reversed, given the use of mod and hash operations.
> >
> > > GPT-based architectures like Llama or Mistral
> >
> > Llama and Mistral are not GPT-based architectures... GPT, Llama, and Mistral are all decoder-only architectures, known for their strong generative capabilities. I am curious how their comprehension performance compares to encoder-decoder architectures, like T5. And the authors did not clearly justify the choice of T5 over decoder-only architectures.
> >
> > > does not require external alignment between different modalities
> >
> > This statement is inaccurate. The training of the 3D embedding layer alongside the 1D embedding layer involves alignment between the two modalities.

---

> ### Author Response · Authors · 2024-11-25
> **Further Comments by Authors (1/2)**
>
> Thanks for your further response. We want to further clarify several points and hope these can address your concerns.
>
> > The authors repeatedly claim in the rebuttal and manuscript that the proposed model is "unified." However, I do not agree with this characterization. In the context of multi-modal LLMs, "unified" typically refers to unifying comprehension and generation capabilities. Your model does not support the generation of 3D tokens, so it cannot truly be considered "unified." Instead, it would be more accurate to state that your model uses 3D tokens in a discrete form.
> >
>
> Our characterization of the model as "unified" primarily refers to
>
> (1) the ability of our approach to **jointly process 1D molecular sequences, 3D molecular structures, and text** within a single framework, enabling comprehensive learning across all of them. Furthermore,
>
> (2) our model is **pre-trained on tasks involving these related modalities**, ensuring throughout interactions between 1D, 3D, and textual information during pre-training, rather than requiring separate pre-training stages for each modality.
>
> (3) We also emphasize unified as it **does not rely on external encoders** for the different modalities, which distinguishes it from approaches that require external 3D encoders or other specialized components.
>
>
> The reviewer repeatedly challenge the ‘unified’ must contain comprehension and generation capabilities.
>
> (1) We refer GPT’s answer to this question: “In summary, a "unified model" in deep learning generally refers to a versatile, multitasking model that can handle a variety of tasks with a single underlying architecture. While many such models are capable of both understanding and generation, being "unified" does not inherently require competence in both areas. The unified nature often refers more to the model's architectural flexibility and task adaptability rather than specific capabilities in understanding and generation.”
>
> (2) If the reviewer insists this should be modified, **we are open to revise the claim as we responded before**. Please kindly reconsider the contribution of our method/framework instead of only the term characterization.
>
> > An unbounded representation space should not necessarily be viewed as a disadvantage. It allows for more diverse representations compared to a finite token space and often results in better performance than discrete architectures in comprehension tasks. The primary purpose of discretization is to enable generation, as demonstrated in various multi-modal LLM works. This often involves sacrificing some comprehension performance to achieve generative capabilities. The authors did not clearly justify the choice of the E3FP algorithm over a 3D molecular encoder. Additionally, it is difficult to claim that the discretization process can be reversed, given the use of mod and hash operations.
> >
>
> We do not claim that unbounded representation space is a disadvantage, what we want to say is that discrete representation is more suitable for our framework. The primary motivation for using E3FP lies in its ability to **support the cross modeling of 1D molecular sequences, 3D molecular structures, and text with minimal modifications to the architecture of LLMs**. By transforming 3D molecular structures into discrete tokens, we enable **seamless integration into the pre-training framework of LLMs without the need for external encoders or complex late-stage alignment mechanisms**.
>
> Regarding the reversibility of the discretization process, we agree that the use of mod and hash operations inherently introduces a small probability of collisions. **However, these collisions are extremely rare.** Additionally, we believe that a well-designed reverse model should be capable of reconstructing the overall 3D molecular structure from a sequence of 3D tokens. By leveraging the context provided by the sequence, such a model could resolve potential conflicts arising from rare token collisions.

---

> > ### Author Response · Authors · 2024-11-28
> > **Further Comments by Authors (2/2)**
> >
> > > Llama and Mistral are not GPT-based architectures... GPT, Llama, and Mistral are all decoder-only architectures, known for their strong generative capabilities. I am curious how their comprehension performance compares to encoder-decoder architectures, like T5. And the authors did not clearly justify the choice of T5 over decoder-only architectures.
> > >
> >
> > Several of the baseline models we compare against, **such as MolX [1] and 3D-MoLM[2], are based on decoder-only architectures like Llama. Our results consistently outperform these baselines across various tasks**, which provides some evidence that our approach effectively leverages the T5 architecture for molecular modeling. However, we do not claim that encoder-decoder architectures like T5 are inherently better or worse than decoder-only architectures like GPT, Llama, or Mistral in the molecular domain.
> >
> > The choice of T5 was motivated by its bidirectional attention mechanism, which is well-suited for tasks requiring strong cross-modal interactions, particularly in molecular understanding. Also, followed by previous works like MolT5[3] and nach0[5]. We acknowledge the power of decoder-only architectures, **but we are curious why this could be a weakness point,** **it’s clear that both T5 and decoder-only architectures are valuable and we should allow the voice of different architectures**.
> >
> > > does not require external alignment between different modalities. This statement is inaccurate. The training of the 3D embedding layer alongside the 1D embedding layer involves alignment between the two modalities.
> > >
> >
> > When we refer to **"external alignment"**, we specifically mean the multi-stage alignment process required by **separate architectures**, where different components (e.g., 3D molecule encoder and text decoder) **undergo independent pre-training** before being combined for joint molecule-text alignment. 3D-MolT5 aligns the 1D and 3D embeddings directly during the pre-training process in a single stage, eliminating the need for external alignment or multi-phase training. This streamlined approach reduces complexity and enhances cross-modal interactions from the start.
> >
> > **References**
> >
> > [1] Le, Khiem, et al. "MolX: Enhancing Large Language Models for Molecular Learning with A Multi-Modal Extension." arXiv preprint arXiv:2406.06777 (2024).
> >
> > [2] Li, Sihang, et al. "Towards 3D Molecule-Text Interpretation in Language Models." *The Twelfth International Conference on Learning Representations*.
> >
> > [3] Edwards, Carl, et al. "Translation between Molecules and Natural Language." *Proceedings of the 2022 Conference on Empirical Methods in Natural Language Processing*. 2022.
> >
> > [4] Livne, Micha, et al. "nach0: Multimodal natural and chemical languages foundation model." *Chemical Science* 15.22 (2024): 8380-8389.

---

> ### Comment · Reviewer_NdAP · 2024-11-29
>
> I am not inclined to reject this paper over minor issues. I have read the authors’ initial response and was generally satisfied with it, so I had intended to increase my rating. However, I have not yet had the bandwidth to reply and update my rating. It is unclear to me why the authors decided to revise their initial response and adopt a more aggressive tone. Nonetheless, I appreciate the novelty of the proposed 3D tokens and 1D tokens alignment, and the integration of pre-training denoising and translation tasks. My overall attitude toward this paper remains positive, as reflected in my official review. My comments are meant to provide constructive feedback and suggestions.
>
> > we do not believe this should be a rejection reason
>
> I did not suggest this as a rejection reason but merely shared my opinion. In multi-modal research, “unified” typically refers to unifying understanding and comprehension. It is the authors’ decision whether to follow the conventions.
>
> > with minimal modifications to the architecture of LLMs
>
> This statement is not accurate. In multi-modal LLMs, we typically do not modify the architecture of the LLM itself. Perhaps you meant “with minimal modifications to the architecture of the molecule-text pipeline / framework.”
>
> > complex late-stage alignment mechanisms
>
> I am unclear on the meaning of the term “late-stage alignment.” Perhaps you meant “cross-modal alignment.”
>
> > but we are curious why this could be a weakness point, it’s clear that both T5 and decoder-only architectures are valuable and we should allow the voice of different architectures.
>
> I agree that we should value the diversity of architectures. However, since T5 is the chosen base model, it would strengthen your paper to explain why T5 was selected over other models like Mistral or Llama. Including quantitative comparisons would make your justification more robust and persuasive.
>
>
> During the rebuttal, the authors made some errors in basic terminology, such as referring to a “GPT-based model” when they likely meant “decoder-only model.” And certain statements remain vague. Refining these in the paper would make the work more accessible to readers.
> Today is Thanksgiving, a day for kindness. Let us work together to make the community better.

---

> > ### Author Response · Authors · 2024-11-29
> >
> > Thanks for your thoughtful and constructive feedback. We are pleased that our response help address your concerns and we appreciate your willingness to increase your rating. The reason of revising the first version of the response is that, after reviewing our previous response, we found it didn’t put much information for reviewer and other readers to have a valuable and thoughtful discussion about the response. We hope to provide more detailed statements and our insights on some details. Follow your last kind response, here are some notes:
> >
> > - We appreciate your suggestions on the use of “unified”, and we will consider revising in the manuscript to better align with the conventions of the field and make our intent clearer.
> > - We do give some imprecise working about the framework in the initial response. What we intended to convey is that **3D-MolT5 requires minimal modifications to the architecture of the molecule-text framework**, rather than the language model architecture itself. We appreciate your understanding.
> > - Yes, the late-stage alignment is what you mentioned. We indeed meant **cross-modal alignment** between external encoder and language decoder as done in other works.
> > - Thanks for your suggestion to let us put more explanations about why we choose T5. We will include the discussions as we respond in the previous messages in our revised paper, and we have also presented the strong results compared to decoder-only models. We will highlight the results and the reasons as you suggested. Besides, we intend to give more quantitative and comprehensive comparison in the final version.
> >
> > We apologize if the tone of our previous response seemed overly strong. That was never our intention, and we truly value your constructive feedback. We appreciate your commitment to improving the community, and we share that goal.
> >
> > On this Thanksgiving Day, we wish you a joyful holiday :) and remain grateful for your efforts in helping make the research community better. We will continue working to improve our work and contribute to the field.

---

> > > ### Comment · Reviewer_NdAP · 2024-12-01
> > >
> > > Thank you for your response. I will raise my rating from 5 to 6. Here are some suggestions and concerns from my side that I hope the authors can address in the final version:
> > >
> > > - Including a primary figure illustrating how the model works would enhance understanding for the readers.
> > >
> > > - The model’s inability to achieve 3D structure generation is due to multiple factors, not just collisions from hash functions and modulus operations. **The authors have not been fully transparent about this.** There are several pooling operations in the algorithm, such as pooling embeddings for each iteration into the 3D embedding and pooling the 1D and 3D embeddings. Additionally, hash functions are widely known to be non-invertible. You cannot decode an integer into a list and then reconstruct the atoms. **This limitation should be explicitly acknowledged in the paper, rather than described vaguely as “left for future work.”**

---

> ### Author Response · Authors · 2024-12-01
>
> Thank you for your feedback and for increasing the score. Following your suggestions, we will include a primary figure in the final version to better illustrate how the model works. Regarding the limitations in 3D structure generation, we agree that these challenges need to be more explicitly acknowledged. We will revise the manuscript to provide a clearer explanation of the underlying difficulties and ensure that these points are transparently discussed.

---

### Official Review · Reviewer_7qiv · 2024-11-01

**Soundness:** 2
**Presentation:** 3
**Contribution:** 3
**Rating:** 6
**Confidence:** 5

**Summary:**

The authors introduce a novel approach that integrates spatial molecular data by calculating a set of E3FP-based discrete tokens for each atom, then combining these tokens with corresponding SELFIES embeddings. Additionally, they propose a new pretraining scheme designed to improve the alignment and interaction between structural (1D) and spatial (3D) tokens. Finally, the authors present a thorough evaluation of the 3DMol-T5 model, which incorporates these novel methods into an encoder-decoder architecture, across various molecular tasks.

**Strengths:**

The paper is well-written and easy to follow, with supplementary materials providing all necessary details to reproduce the method. The concept of integrating spatial molecular fingerprints alongside molecular sequence representations is a novel contribution in the field of language models. Experimental results demonstrate a notable improvement in model performance compared to baseline models.

**Weaknesses:**

Despite the comprehensive benchmarking of the model across various molecular tasks, the limited ablation studies make it difficult to assess the individual contributions of each component of the proposed approach to overall model performance. Additionally, a minor drawback is the absence of non-language model baselines in the Molecular Property Prediction tasks section. Further details on these points are provided in the Questions section.

**Questions:**

As noted in the Weaknesses section, although the proposed method achieves significant performance improvements over baseline models, it appears overly complex and includes numerous subcomponents. The necessity of each part is not sufficiently justified.
* The authors introduce a novel pretraining scheme comprising five subtasks. Are all these subtasks equally essential for optimal model performance, or could the model show similar performance using only one or two? Intuitively, the “3D molecule to text translation” task might be sufficient, as it encompasses all three modalities—text, 1D molecular structure, and 3D molecular structure. It would be valuable to explore this question further by showing the performance of 3DMol-T5 after pretraining on each individual task while excluding the remaining four.
* Several recent works [a, b] have demonstrated that language models can process spatial molecular data directly as text containing atom coordinates. Compared to the proposed E3FP-based discrete tokens, this direct coordinate representation is considerably simpler, more transparent, and requires no model modifications. Furthermore, it can serve as both input and output (for example, in spatial molecular generation tasks). Providing a comparison with this approach is important to justify the use of the E3FP-based discrete token scheme.
* The proposed alignment scheme for combining 1D and 3D molecular information involves summing their embeddings as $E=\frac{1}{2}E_{1D} + \frac{1}{2}E_{3D}$ which necessitates modifications to the language model's source code—requiring time and substantial programming knowledge to adapt it for other models. However, is this modification truly necessary? Could the E3FP tokens be interpreted as textual tokens and used directly within the text without altering the model? Intuitively, a sequence such as “$s_0, d_{01}, d_{02}, d_{03}, s_1, d_{11}, d_{12}, d_{13} \cdots$” might accomplish this. It would be useful to elaborate on this question.

Minor suggestions:
* While the QM9 dataset is suitable for demonstrating the model's capability in using 3D molecular information to predict quantum properties, I recommend that the authors consider the QMUGS dataset [c] due to the greater diversity in molecular structures and quantum properties. Notably, QMUGS includes atom-wise properties, which could provide a more comprehensive evaluation. Testing model performance on this dataset could enhance the significance of the experimental section.

a. Language models can generate molecules, materials, and protein binding sites directly in three dimensions as XYZ, CIF, and PDB files, 2023

b. BindGPT: A Scalable Framework for 3D Molecular Design via Language Modeling and Reinforcement Learning, 2024

c. QMugs, quantum mechanical properties of drug-like molecules, 2022

---

> ### Author Response · Authors · 2024-11-19
>
> Thanks for your insightful comments. We’d like to address your concerns as follows.
> > W1: Despite the comprehensive benchmarking of the model across various molecular tasks, the limited ablation studies make it difficult to assess the individual contributions of each component of the proposed approach to overall model performance. Additionally, a minor drawback is the absence of non-language model baselines in the Molecular Property Prediction tasks section. **Further details on these points are provided in the Questions section.**
>
> - For more ablation studies about pre-training task, we provide a thorough analysis of the individual contributions of each component of our proposed approach and their impact on overall model performance. Please kindly refer to the **response to Q1**.
> - As for non-language model baselines in the Molecular Property Prediction tasks, we would like to clarify that **Uni-Mol**[1] is a non-language model baseline. Following your suggestion, we conducted additional experiments on the MoleculeNet[3] dataset to provide a broader range of **non-language model baselines** (e.g., GNN-based models) and other LM **baselines**. As shown in the following Table 1, 3D-MolT5 achieves superior performance across all tasks, demonstrating its effectiveness in leveraging 3D molecular information in a unified framework.
>
> We appreciate your feedback and have incorporated these additional results into **Appendix B** and **Table 7** (**Page 19-20**) in the revised manuscript to better highlight the contributions and versatility of 3D-MolT5.
>
> Table 1: Results (AUROC) for the molecular property prediction task on MoleculeNet[3].
>
> ||BACE|BBBP|HIV|ClinTox|Avg|
> |---|---|---|---|---|---|
> |Non-LMbased||||||
> |GraphCL|75.4|69.7|78.5|76.0|74.9|
> |GraphMVP-C|81.2|72.4|77.0|77.5|77.0|
> |MGSSL|79.7|70.5|79.5|80.7|77.6|
> |MolCLR|**89.0**|73.8|80.6|93.2|84.2|
> |GEM|85.6|72.4|80.6|90.1|82.2|
> |Uni-Mol|85.7|72.9|**80.8**|91.9|82.8|
> |LM-based||||||
> |KV-PLM|71.9|66.9|68.8|84.3|73.0|
> |MoMu|76.7|70.5|75.9|79.9|75.8|
> |MolFM|83.9|72.9|78.8|79.7|78.8|
> |Galactica|72.7|59.6|75.9|82.2|72.6|
> |UniMoT|83.7|71.4|78.5|92.9|81.6|
> |3D-MolT5|88.1|**76.5**|**80.8**|**95.4**|**85.2**|
>
> > Q1: Are all these subtasks equally essential for optimal model performance, or could the model show similar performance using only one or two? Intuitively, the “3D molecule to text translation” task might be sufficient, as it encompasses all three modalities—text, 1D molecular structure, and 3D molecular structure. It would be valuable to explore this question further by showing the performance of 3DMol-T5 after pretraining on each individual task while excluding the remaining four.
>
> Following your suggestion, we pre-train 3D-MolT5 on each individual subtask while excluding the remaining four. We evaluate on the **3D molecule to text translation** task (as this task involves all modalities). The results are shown in the following table, the observations are:
>
> - The "3D molecule to text translation" task provides a strong foundation and achieves competitive results, as it inherently involves all three modalities (text, 1D molecular sequnece, and 3D molecular structure).
> - "1D denoising" also significantly enhances performance, highlighting that pre-training on 1D molecular sequences provides a crucial foundation for understanding 3D molecular structures.
> - Certain pre-training tasks, such as "1D + 3D joint denoising", "3D to 1D translation", and "text to 1D molecule translation", resulted in suboptimal performance. This is expected since the **significant gap exists between the objectives of these pre-training tasks and the downstream task.** The downstream task requires the input of 3D molecular structures and the output of text descriptions, whereas "1D + 3D joint denoising" and "3D to 1D translation" do not involve any textual pre-training. Moreover, for "3D to 1D translation", the input and output spaces are reversed compared to the downstream task, which might contribute to the misalignment in learned representations.
> - **The joint pre-training on all five subtasks yields the best performance, demonstrating the complementary benefits of multi-task pre-training.**
>
> We have included these additional ablations in **Appendix C** (**Page 20-21**) in the revised version of manuscript.
>
> Table 2: Ablation results for each type of pre-training tasks.
>
> | |BLEU-2 ↑| BLEU-4 ↑| ROUGE-1 ↑ | ROUGE-2 ↑ | ROUGE-L ↑ | METEOR ↑ |
> |---|---|---|---|---|---|---|
> |only 1D denoising| 39.78 | 31.29 | 46.11 | 30.48 | 40.31 | 41.94 |
> |only 1D+3D joint denoising| 30.76 | 22.11 | 37.41 | 21.20 | 31.71 | 32.84 |
> |only 3D to 1D translation| 29.66 | 21.03 | 36.08 | 20.02 | 30.44 | 31.51 |
> |only 3D molecule to text translation| 40.17 | 32.09 | 45.94 | 30.56 | 40.10 | 42.16 |
> |only text to 1D molecule translation| 32.15 | 23.50 | 39.86 | 23.81 | 34.51 | 35.50 |
> |3D-MolT5| **42.05** | **34.16** | **48.13** | **33.20** | **42.33** | **44.69** |

---

> ### Author Response · Authors · 2024-11-19
>
> > Q2: Several recent works [4, 5] have demonstrated that language models can process spatial molecular data directly as text containing atom coordinates. Compared to the proposed E3FP-based discrete tokens, this direct coordinate representation is considerably simpler, more transparent, and requires no model modifications. Furthermore, it can serve as both input and output (for example, in spatial molecular generation tasks). Providing a comparison with this approach is important to justify the use of the E3FP-based discrete token scheme.
> >
>
> Thanks for the comment. We agree that directly encoding spatial molecular data as text containing atom coordinates is a simpler approach. However, our primary focus is on 3D molecular understanding tasks, such as property prediction and captioning, we believe the E3FP-based[7] discrete token scheme offers significant advantages:
>
> 1. **Input Length and Computational Efficiency**
>
>     Representing spatial coordinates directly as text substantially increases input sequence length, especially when dealing with large molecules. This not only introduces additional computational overhead considering the quadratic complexity of the attention mechanism, but also complicates the model's learning process, as longer sequences can dilute meaningful patterns within the data. In contrast, by encoding 3D structure into 3D tokens and aligning 1D and 3D embeddings at the atomic level, our approach maintains a balanced and scalable representation while avoiding unnecessary computational complexity.
>
> 2. **Semantic Representation of Numerical Data**
>
>     Tokenizing coordinates as text often results in a loss of numerical semantic relationships. For instance, the tokens for the numbers "123" and "124" are treated as entirely distinct, despite their numerical proximity. This lack of semantic similarity makes it challenging for the model to capture meaningful numerical relationships[6], such as proximity or continuity. In contrast, the E3FP algorithm encodes 3D molecular structures as discrete tokens based on hierarchical and spatial substructures, preserving critical spatial relationships in a form that is more interpretable and useful for the model.
>
> 3. **Lack of SE(3)-Invariance**
>
>     Representing spatial data directly as coordinates can struggle with preserving SE(3)-Invariance, i.e., invariance to molecular rotations, translations, and reflections. Without explicit adjustments, such representations may lead to inconsistencies in encoding the same molecule under different orientations. But E3FP is inherently SE(3)-invariant (discussed in Appendix A.2), ensuring that the discrete tokens remain consistent regardless of molecular orientation, which is crucial for tasks like 3D molecular understanding.
>
>
> We recognize that 3D-MolT5 currently does not support direct generation of 3D molecular structures or coordinates. This is an area we plan to explore in future work. As suggested in your Q3, integrating both 1D and E3FP-based 3D tokens into a unified sequence and incorporating an external decoder to reconstruct 3D molecular structures from the generated tokens would be a promising direction. This would not only address spatial molecular generation tasks but also provide a comprehensive framework for both understanding and generation in molecular modeling. We appreciate your suggestion and have included a discussion of this comparison and potential future directions in **Appendix D.1** (**Page 21-22**) in the revised manuscript to highlight these considerations.

---

> ### Author Response · Authors · 2024-11-19
>
> > Q3: The proposed alignment scheme for combining 1D and 3D molecular information involves summing their embeddings as  which necessitates modifications to the language model's source code—requiring time and substantial programming knowledge to adapt it for other models. However, is this modification truly necessary? Could the E3FP tokens be interpreted as textual tokens and used directly within the text without altering the model? Intuitively, a sequence such as $s_0,d_{01},d_{02},d_{03},s_1,d_{11},d_{12},d_{13}...$ might accomplish this. It would be useful to elaborate on this question.
> >
> - Thanks for this insightful question. Following your suggestion, we conducted preliminary experiments on the **3D molecule to text translation** task using the PubChem[2] dataset. Specifically, we compared our summing 1D and 3D embeddings method with your suggestion, where the E3FP tokens are directly concatenated with the 1D molecular tokens in a sequential manner. Both methods were trained from scratch for a fair comparison, and the results are presented in the following Table 3.
> - As expected, the two approaches achieved comparable performance and your suggestion is valuable. However, it is worth noting that the sequential approach significantly increases the input sequence length, particularly for larger molecules, which lead to higher computational costs and longer training times (in our experiments, it take more than 1.5 times as long to converge as embedding summation setting). Despite this drawback, we are interested to modify our framework in your suggested way, since this method does not require modifications to the model's source code and aligns more naturally with tasks involving 3D molecular generation.
>
> We appreciate your suggestion and have included these results and discussions in **Appendix D.2** and **Table 10** (**Page 22**) in the revised manuscript to highlight the potential for future work.
>
> Table 3: Results comparison between sum of 1D and 3D embedding versus sequential concatenation of 1D and 3D tokens.
>
> |  | BLEU-2 ↑ | BLEU-4 ↑ | ROUGE-1 ↑ | ROUGE-2 ↑ | ROUGE-L ↑ | METEOR ↑ |
> | --- | --- | --- | --- | --- | --- | --- |
> | Embedding Summation | 37.78 | 29.61 | 42.94 | 27.51 | 37.40 | 38.86 |
> | Sequential Concatenation | 37.65 | 29.34 | 43.25 | 27.64 | 37.45 | 38.98 |
>
> > Minior suggestions: While the QM9 dataset is suitable for demonstrating the model's capability in using 3D molecular information to predict quantum properties, I recommend that the authors consider the QMUGS dataset [9] due to the greater diversity in molecular structures and quantum properties. Notably, QMUGS includes atom-wise properties, which could provide a more comprehensive evaluation. Testing model performance on this dataset could enhance the significance of the experimental section.
> >
>
> Thanks for your suggestion. In this work, to ensure a fair comparison with related LM-based methods, we evaluated our model on widely used benchmarks, such as PubChemQC[2], PubChem[2], and QM9[8] datasets.
>
> We recognize the value of the QMUGS[9] dataset. However, due to time constraints, we may not be able to include experiments on QMUGS during rebuttal. We appreciate your recommendation and plan to involve QMUGS in the future.
>
> **References**
>
> [1] Zhou, Gengmo, et al. "Uni-Mol: A Universal 3D Molecular Representation Learning Framework." *The Eleventh International Conference on Learning Representations*.
>
> [2] Li, Sihang, et al. "Towards 3D Molecule-Text Interpretation in Language Models." *The Twelfth International Conference on Learning Representations*.
>
> [3] Wu, Zhenqin, et al. "MoleculeNet: a benchmark for molecular machine learning." *Chemical science* 9.2 (2018): 513-530.
>
> [4] Flam-Shepherd, Daniel, and Alán Aspuru-Guzik. "Language models can generate molecules, materials, and protein binding sites directly in three dimensions as xyz, cif, and pdb files." *arXiv preprint arXiv:2305.05708* (2023).
>
> [5] Zholus, Artem, et al. "BindGPT: A Scalable Framework for 3D Molecular Design via Language Modeling and Reinforcement Learning." *arXiv preprint arXiv:2406.03686* (2024).
>
> [6] https://community.openai.com/t/why-9-11-is-larger-than-9-9-incredible/869824/5
>
> [7] Axen, Seth D., et al. "A simple representation of three-dimensional molecular structure." Journal of medicinal chemistry (2017).
>
> [8] Fang, Yin, et al. "Mol-Instructions: A Large-Scale Biomolecular Instruction Dataset for Large Language Models." *The Twelfth International Conference on Learning Representations*.
>
> [9] Isert, Clemens, et al. "QMugs, quantum mechanical properties of drug-like molecules." *Scientific Data* 9.1 (2022): 273.

---

> ### Comment · Reviewer_7qiv · 2024-11-25
>
> I appreciate the authors' detailed responses to my questions and the additional ablation study experiments provided to justify their approach. I'm pleased to see that my feedback contributed to strengthening the research. I hope the final version will incorporate all the findings, including the insights on the importance of the pretraining task and the merging schemes for SELFIES and 3D fingerprints. I have decided to raise my score.

---

> > ### Author Response · Authors · 2024-11-26
> >
> > Thanks for your further response. We appreciate your recognition of our efforts and are glad the additional experiments addressed your concerns. We will include the findings in the final version accordingly.

---

### Official Review · Reviewer_GGJY · 2024-11-04

**Soundness:** 3
**Presentation:** 3
**Contribution:** 3
**Rating:** 6
**Confidence:** 3

**Summary:**

This paper introduces 3D-MolT5, a novel framework for integrating molecular sequence and 3D structural information within a unified language model. While recent language models have advanced molecular science applications, most neglect critical 3D structural data essential for accurately understanding molecular properties. To address this, 3D-MolT5 uses a unique 3D token vocabulary that captures detailed 3D substructure information, enabling it to encode molecular sequences, spatial structures, and natural language together in a cohesive architecture. Through multi-task pre-training, 3D-MolT5 develops robust cross-modal comprehension. Experimental results show strong generalization, with the model achieving nearly a 70% improvement in molecular property prediction compared to current methods.

**Strengths:**

This work shows strength in its comprehensive comparison with various existing models, showcasing 3D-MolT5’s effectiveness across different approaches and highlighting its strong performance in molecular structure understanding. By integrating both molecular sequence and 3D structural information, the study addresses a crucial aspect of molecular modeling that is often underexplored, underscoring its relevance in capturing complex molecular properties.

**Weaknesses:**

While the work presents a solid approach, the concept of multimodal integration itself is not entirely new. Including a comparison with fine-tuned large language models (LLMs) that incorporate multimodal data[1] would further strengthen the evaluation, providing a clearer benchmark comparing to a boarder types of multimodal molecular representation models and other LLM as well e.g. Galactica[2].

[1] Le, Khiem, et al. "MolX: Enhancing Large Language Models for Molecular Learning with A Multi-Modal Extension." arXiv preprint arXiv:2406.06777 (2024).
[2] Taylor, Ross, et al. "Galactica: A large language model for science." arXiv preprint arXiv:2211.09085 (2022).

**Questions:**

Given that multimodal approaches are increasingly common, how does 3D-MolT5 specifically leverage 3D structural data differently or more effectively than existing fine-tuned LLMs that incorporate multimodal information?
In the model’s tokenized format for 3D structures, what considerations were made for maintaining structural fidelity across different molecule sizes and complexities? How does the approach handle conformational flexibility within molecular ensembles?
Could the model’s performance be further validated on additional downstream tasks, such as reaction prediction or drug-target interaction, to assess its versatility across a broader range of applications in molecular science?

---

> ### Author Response · Authors · 2024-11-19
>
> Thanks for your comments. We’d like to address your concerns as follows.
> > W1.1: While the work presents a solid approach, the concept of multimodal integration itself is not entirely new.
> >
>
> We’d like to clarify that we are not claiming the concept of multimodal integration, we emphasize that our primary motivation is not solely to present another multimodal integration framework but to highlight the significance of incorporating **3D molecular information** into language models in a **unified** manner. Here is a more detailed clarification of our motivation and contributions:
>
> 1. **Significance of 3D Information**
>
>     While prior works have demonstrated success in molecule-text multimodal modeling, they mainly focus on 1D molecular sequences (e.g., SMILES, SELFIES)[2-4] or 2D molecular graphs[1,5], the **3D molecular structures remain underexplored**[6]. However, the 3D molecular structures play critical role in determining molecular functions and properties[17]. Our work specifically addresses this gap by proposing a novel framework that emphasizes the importance of 3D molecular information.
>
> 2. **Unified Integration of 1D molecular sequence, 3D molecular structure, and text**
>
>     Unlike existing multimodal approaches that often treat different modalities separately (i.e., molecules are encoded using separate encoders[5-6]), we propose a **unified integration** strategy. By introducing discrete tokenization of 3D molecular structures (which is novel), we seamlessly incorporate 1D molecular sequence, 3D molecular structure, and text modalities into a single model. This integration not only **simplifies the framework** but also **ensures extensive cross-modal interaction during pre-training**, enhancing the representation and alignment across modalities. To our knowledge, our unified framework is the first in the molecular domain to model 3D molecular information in this tokenized way, as existing works[2-4] primarily focus on integrating 1D or 2D modalities.
>
> > W1.2: Including a comparison with fine-tuned large language models (LLMs) that incorporate multimodal data[1] would further strengthen the evaluation, providing a clearer benchmark comparing to a boarder types of multimodal molecular representation models and other LLM as well e.g. Galactica[2].
> >
>
> Thanks for your insightful comments. We want to emphasize that the main motivation is to **integrate 3D molecular information** in a unified framework instead of only multimodal learning. Similar to previous works, [1] only introduces 2D molecular information. We agree with comparing with more multimodal molecular representation models. Indeed, we have already compared with several fine-tuned LLMs that incorporate multimodal data, such as **3D-MoLM**[6], **InstructMol-GS**[7], **MoMu**[8], and **GIT-Mol**[9] as detailed in our paper. According to your suggestion, we further compared with **MolX**[1], **Galactica**[2], and other baselines on 3D molecule captioning, retrosynthesis (newly added benchmark), and molecular property prediction tasks.
>
> - **Comparison with MolX.** We evaluate 3D molecule captioning on the PubChem[6] dataset and retrosynthesis on the USPTO-50k dataset[1]. Results in the following Table 1 and 2 show that **3D-MolT5 achieves much better performance than MolX on both tasks**, particularly on 3D molecule-to-text translation, where incorporating 3D information enables more accurate and detailed descriptions.
>
> Table 1: Results for the 3D molecule captioning task on PubChem[6] dataset.
>
> |  | BLEU-2 ↑ | BLEU-4 ↑ | ROUGE-1 ↑ | ROUGE-2 ↑ | ROUGE-L ↑ | METEOR ↑ |
> | --- | --- | --- | --- | --- | --- | --- |
> | MolT5 | 25.87  | 17.28 | 34.07 | 16.42  | 23.41  | 28.04 |
> | LlaSMol | 26.71 | 18.06 | 38.75 | 22.77 | 33.32 | 32.63 |
> | 3D-MoLM | 29.82 | 22.39 | 39.12 | 23.62 | 32.64 | 34.34 |
> | MolX | 31.40 | 24.25 | 44.20 | 28.96 | 38.76 | 39.55 |
> | 3D-MolT5 | **42.05** | **34.16** | **48.13** | **33.20** | **42.33** | **44.69** |
>
> Table 2: Results for the retrosynthesis task on USPTO-50k[1] dataset.
>
> |  | BLEU-2 ↑ | Levenshtein ↓ | RDKit FTS ↑ | Validity ↑ |
> | --- | --- | --- | --- | --- |
> | ReactionT5 | 81.63 | 17.69 | 0.7400 | 97.58 |
> | LlaSMol-7B | 50.09 | 31.28 | 0.7351 | 99.65 |
> | 3D-MoLM | 81.31 | 16.21 | 0.7341 | 90.31 |
> | MolX | 82.59 | 15.74 | 0.7466 | 92.19 |
> | 3D-MolT5 | **86.23** | **14.08** | **0.7538** | **100.00** |

---

> ### Author Response · Authors · 2024-11-19
>
> - **Comparison with Galactica.** We compared molecular property prediction performance on the MoleculeNet[12] benchmark. Results in Table 3 show that **3D-MolT5 significantly outperforms Galactica**, which lacks 3D molecular representation. The inclusion of 3D tokenization in our framework allows for a more comprehensive understanding of the spatial and chemical properties of molecules, leading to superior performance on such tasks.
>
>     Table 3: Results (AUROC) for the molecular property prediction task on MoleculeNet[12].
>
>     |  | BACE | BBBP | HIV | ClinTox | Avg |
>     | --- | --- | --- | --- | --- | --- |
>     | MolCLR | **89.0** | 73.8 | 80.6 | 93.2 | 84.2 |
>     | Uni-Mol | 85.7 | 72.9 | **80.8** | 91.9 | 82.8 |
>     | MolFM | 83.9 | 72.9 | 78.8 | 79.7 | 78.8 |
>     | Galactica-6.7B | 58.4 | 53.5 | 72.2 | 78.4 | 65.6 |
>     | Galactica-30B | 72.7 | 59.6 | 75.9 | 82.2 | 72.6 |
>     | Galactica-120B | 61.7 | 66.1 | 74.5 | 82.6 | 71.2 |
>     | UniMoT | 83.7 | 71.4 | 78.5 | 92.9 | 81.6 |
>     | 3D-MolT5 | 88.1 | **76.5** | **80.8** | **95.4** | **85.2** |
>
>
> These results further demonstrate the advantages of 3D-MolT5's unified integration of 1D molecular sequence, 3D molecular structure, and text over existing multimodal LLMs.
>
> We appreciate your suggestion and have added them to related works and incorporated the complete version of additional comparisons into the **Table 4** (**Page 9**), **Appendix B**, and **Table 8-9** (**Page 19-21**) in the revised manuscript.
>
> > Q1: Given that multimodal approaches are increasingly common, how does 3D-MolT5 specifically leverage 3D structural data differently or more effectively than existing fine-tuned LLMs that incorporate multimodal information?
> >
>
> Thank you for raising this question. We provide a concise response highlighting the core differences and effectivenesses of 3D-MolT5 as follows:
>
> - **Limitations of separate modeling in existing methods.** While multimodal approaches are indeed becoming more prevalent in molecule-text modeling, the prevailing methods[1,5-9] **separately model each modality using dedicated encoders and then aligning their represent spaces**. For instance, in 3D-MoLM[6], molecules are encoded through an external 3D encoder Uni-Mol[15], which is subsequently aligned with a language decoder Llama-2[16] to integrate multimodal information in a joint representation space. This approach of leveraging separate models for different modalities has inherent limitations.:
>     - **Requiring additional components and processes**, such as projection layers and alignment mechanisms, to ensure compatibility between the representations of different modalities. Such late-stage alignment processes are challenging to optimize effectively.
>     - The alignment is further difficult particularly when the **paired data between text and 3D structures are limited**.
>     - Treating modalities separately **hinders early and rich cross-modal interactions**, which are critical for enhancing joint representations and ensuring a seamless fusion of information.
> - 3D-MolT5 adopts a fundamentally different approach by introducing **discrete tokenization of 3D molecular structures**. Discretizing the 3D molecule structures into tokens is a pretty novel attempt and a critical contribution. This allows us to model 1D molecular sequences, 3D molecular structures, and text within a **unified architecture**. By transforming 3D information into discrete tokens compatible with the pre-training framework of LLMs, we **seamlessly integrate these modalities without the need for external encoders or late-stage alignment**. This unified framework simplifies the architecture while **enhancing cross-modal interactions during pre-training**, enabling 3D-MolT5 to effectively leverage 3D structural data for downstream tasks.
>
> In a summary, tokenizing 3D molecular structures and our novel integration strategy are two key innovations of our work and addresses the limitations of separate modeling in existing multimodal approaches. As demonstrated in our experiments, 3D-MolT5 achieves superior performance across molecule-text tasks, underscoring the value of unifying 1D, 3D molecular, and textual modalities in a single coherent model.

---

> > ### Comment · Reviewer_GGJY · 2024-11-27
> >
> > Thank you for addressing my feedback and conducting additional experiments. While the performance improvement remains not significantly higher and the novelty is limited, I recognize the effort put into refining the work. Based on this, I have updated my score to 6.

---

> > > ### Author Response · Authors · 2024-11-28
> > >
> > > Thanks for your thoughtful feedback and acknowledgment. We believe that the refinements contribute meaningfully to the overall work. Your updated score is much appreciated.

---

> ### Author Response · Authors · 2024-11-19
>
> > Q2: In the model’s tokenized format for 3D structures, what considerations were made for maintaining structural fidelity across different molecule sizes and complexities? How does the approach handle conformational flexibility within molecular ensembles?
> >
> - Our approach leverages the E3FP[13] algorithm to tokenize 3D structures, which **progressively encodes molecular structures through iterative expansions of the encoding radius**. This iterative process allows the model to adapt to molecules of varying sizes and complexities. Specifically, at each iteration, the radius is expanded by 1.718 Å, and in the 3D-MolT5 framework, we set the maximum iteration count to 3, enabling the encoding of molecular substructures within a radius of 5.154 Å. This configuration effectively accommodates the majority of molecules in standard datasets. For larger molecules, the number of iterations can be increased to handle greater radii, ensuring the **flexibility to encode larger and more complex molecular structures**.
> - To address **conformational flexibility**, the E3FP algorithm employs MurmurHash3[14] algorithm, which has an extremely low collision rate, ensuring that different substructures, even those with subtle variations, are mapped to distinct 3D tokens. **As detailed in Appendix A.6 and illustrated in Figure 5**, E3FP can effectively capture nuanced differences between conformers, such as slight changes in substituent orientations, by generating distinct tokens for each conformer. This capability ensures that the tokenized representation preserves **structural fidelity and remains sensitive to conformational variations**, enabling the model to handle flexible molecular structures with high accuracy.
>
> > Q3: Could the model’s performance be further validated on additional downstream tasks, such as reaction prediction or drug-target interaction, to assess its versatility across a broader range of applications in molecular science?
> >
> - Thanks for this valuable suggestion. As mentioned in our response to W1.2, we have already validated the performance of 3D-MolT5 on additional downstream tasks, including **retrosynthesis** on the USPTO-50k dataset and **molecular property prediction** on the MoleculeNet dataset. These evaluations demonstrate the model's strong capabilities across a variety of molecular tasks.
> - Following your suggestion, we further assessed the versatility of 3D-MolT5 by testing it on **forward reaction prediction** and **reagent prediction** tasks using the Mol-Instructions[18] dataset. The results, presented in Table 4 and 5, show that 3D-MolT5 consistently outperforms baseline methods on both tasks. These findings highlight the effectiveness of our unified framework in capturing complex molecular interactions and adapting to diverse molecular tasks, supporting the model’s versatility across a broader range of applications.
>
> We appreciate your feedback and have included these new results in the revised version of our manuscript in **Appendix B** and **Table 6** (**Page 19-21**).
>
> Table 4: Results for the forward reaction prediction task on Mol-Instructions[18].
>
> |  | Exact Match ↑ | BLEU ↑ | Levenshtein ↓ | RDK FTS ↑ | MACCS FTS ↑ | Morgan FTS ↑ | Validity ↑ |
> | --- | --- | --- | --- | --- | --- | --- | --- |
> | Mol-Instructions | 0.045 | 0.654 | 27.262 | 0.313 | 0.509 | 0.262 | 1.000 |
> | InstructMol | 0.536 | 0.967 | 10.851 | 0.776 | 0.878 | 0.741 | 1.000 |
> | UniMoT | 0.611 | 0.980 | 8.297 | 0.836 | 0.911 | 0.807 | 1.000 |
> | 3D-MolT5 | **0.879** | **0.994** | **2.842** | **0.955** | **0.978** | **0.942** | 1.000 |
>
> Table 5: Results for the reagent prediction task on Mol-Instructions[18].
>
> |  | Exact Match ↑ | BLEU ↑ | Levenshtein ↓ | RDK FTS ↑ | MACCS FTS ↑ | Morgan FTS ↑ | Validity ↑ |
> | --- | --- | --- | --- | --- | --- | --- | --- |
> | Mol-Instructions | 0.044 | 0.224 | 23.167 | 0.237 | 0.364 | 0.213 | 1.000 |
> | InstructMol | 0.129 | 0.610 | 19.644 | 0.444 | 0.539 | 0.400 | 1.000 |
> | UniMoT | 0.167 | 0.728 | 14.588 | 0.549 | 0.621 | 0.507 | 1.000 |
> | 3D-MolT5 | **0.277** | **0.756** | **13.173** | **0.571** | **0.646** | **0.540** | 1.000 |

---

> > ### Author Response · Authors · 2024-11-19
> >
> > **References**
> >
> > [1] Le, Khiem, et al. "MolX: Enhancing Large Language Models for Molecular Learning with A Multi-Modal Extension." arXiv preprint arXiv:2406.06777 (2024).
> >
> > [2] Taylor, Ross, et al. "Galactica: A large language model for science." arXiv preprint arXiv:2211.09085 (2022).
> >
> > [3] Edwards, Carl, et al. "Translation between Molecules and Natural Language." *Proceedings of the 2022 Conference on Empirical Methods in Natural Language Processing*. 2022.
> >
> > [4] Pei, Qizhi, et al. "BioT5: Enriching Cross-modal Integration in Biology with Chemical Knowledge and Natural Language Associations." *Proceedings of the 2023 Conference on Empirical Methods in Natural Language Processing*. 2023.
> >
> > [5] Liu, Zhiyuan, et al. "MolCA: Molecular Graph-Language Modeling with Cross-Modal Projector and Uni-Modal Adapter." *Proceedings of the 2023 Conference on Empirical Methods in Natural Language Processing*. 2023.
> >
> > [6] Li, Sihang, et al. "Towards 3D Molecule-Text Interpretation in Language Models." *The Twelfth International Conference on Learning Representations*.
> >
> > [7] Cao, He, et al. "Instructmol: Multi-modal integration for building a versatile and reliable molecular assistant in drug discovery." *arXiv preprint arXiv:2311.16208* (2023).
> >
> > [8] Su, Bing, et al. "A molecular multimodal foundation model associating molecule graphs with natural language." *arXiv preprint arXiv:2209.05481* (2022).
> >
> > [9] Liu, Pengfei, et al. "Git-mol: A multi-modal large language model for molecular science with graph, image, and text." *Computers in biology and medicine* 171 (2024): 108073.
> >
> > [10] Yu, Botao, et al. "Llasmol: Advancing large language models for chemistry with a large-scale, comprehensive, high-quality instruction tuning dataset." *arXiv preprint arXiv:2402.09391* (2024).
> >
> > [11] Zhao, Zihan, et al. "Chemdfm: Dialogue foundation model for chemistry." *arXiv preprint arXiv:2401.14818* (2024).
> >
> > [12] Wu, Zhenqin, et al. "MoleculeNet: a benchmark for molecular machine learning." *Chemical science* 9.2 (2018): 513-530.
> >
> > [13] Axen, Seth D., et al. "A simple representation of three-dimensional molecular structure." Journal of medicinal chemistry (2017).
> >
> > [14] Appleby, A. MurmurHash3; GitHub
> >
> > [15] Zhou, Gengmo, et al. "Uni-Mol: A Universal 3D Molecular Representation Learning Framework." *The Eleventh International Conference on Learning Representations*.
> >
> > [16] Touvron, Hugo, et al. "Llama 2: Open foundation and fine-tuned chat models." *arXiv preprint arXiv:2307.09288* (2023).
> >
> > [17] Guo, Zhichun, et al. "Graph-based molecular representation learning." *Proceedings of the Thirty-Second International Joint Conference on Artificial Intelligence*. 2023.
> >
> > [18] Fang, Yin, et al. "Mol-Instructions: A Large-Scale Biomolecular Instruction Dataset for Large Language Models." *The Twelfth International Conference on Learning Representations*.

---

### Author Response · Authors · 2024-11-21
**General Response**

Dear all reviewers:

We sincerely appreciate your valuable time and insightful feedback. A revised version of our manuscript has been uploaded, with changes highlighted in blue for clarity. Below, we provide a summary of the main revisions:

1. **Expanded Benchmark Datasets and Baselines (Appendix B, Table 6-8, Page 19-21)**

    To better validate the effectiveness of our proposed method, we have conducted experiments on a broader range of tasks and benchmark datasets, including:

    - **Forward reaction prediction** on the Mol-Instructions[1] dataset
    - **Reagent prediction** on the Mol-Instructions[1] dataset
    - **Retrosynthesis** on the USTPO-50k[2] and Mol-Instructions[1] datasets
    - **Molecule property prediction** on the MoleculeNet[3] Benchmark

    Additionally, we included more baselines, such as MolX[4], Galactica[5], and UniMoT[6], among others. Results demonstrate that 3D-MolT5 achieves competitive performance not only on 3D-specific tasks but also on 1D and 2D-related tasks, highlighting its versatility and compatibility across diverse molecular modeling tasks and applications.

2. **Additional Discussions and Future Directions (Appendix D, Page 21-22)**

    We have expanded the manuscript to include discussions on:

    - **Comparison with direct coordinate representations**, detailing the advantages of the E3FP-based discrete tokenization framework in computational efficiency, semantic representation, and SE(3)-invariance.
    - **Sequential integration of E3FP and SELFIES tokens**, comparing the performance and analyzing the trade-offs between embedding summation and sequence concatenation approaches.
    - **Future extensions**, exploring possibilities to support 3D molecular generation tasks.

We hope these updates address your concerns and provide a clearer perspective on the contributions and future potential of our work.

**References**

[1] Fang, Yin, et al. "Mol-Instructions: A Large-Scale Biomolecular Instruction Dataset for Large Language Models." *The Twelfth International Conference on Learning Representations*.

[2] Schneider, Nadine, Nikolaus Stiefl, and Gregory A. Landrum. "What’s what: The (nearly) definitive guide to reaction role assignment." *Journal of chemical information and modeling* 56.12 (2016): 2336-2346.

[3] Wu, Zhenqin, et al. "MoleculeNet: a benchmark for molecular machine learning." *Chemical science* 9.2 (2018): 513-530.

[4] Le, Khiem, et al. "MolX: Enhancing Large Language Models for Molecular Learning with A Multi-Modal Extension." arXiv preprint arXiv:2406.06777 (2024).

[5] Taylor, Ross, et al. "Galactica: A large language model for science." arXiv preprint arXiv:2211.09085 (2022).

[6] Zhang, Juzheng, et al. "Unimot: Unified molecule-text language model with discrete token representation." *arXiv preprint arXiv:2408.00863* (2024).

---

### Author Response · Authors · 2024-11-24
**Follow-up on Rebuttal Discussion**

Dear Reviewers,

Thanks for the time, effort, and expertise you have invested in reviewing our paper.

As the deadline for the author-reviewer discussion period is approaching, we are reaching out to you to follow up on our previous rebuttal submission and revised manuscript. We look forward to any further feedback or comments that will help improve our work. If your concerns have been addressed, we would be so much grateful if you could raise your score.

Thanks your time and consideration.

Sincerely,

Authors

---

### Meta-Review · Area_Chair_ZLN1 · 2024-12-19

**Metareview:**

This paper introduces 3D-MolT5, a novel framework that unifies molecular sequence and 3D structural information within a single language model. By mapping fine-grained 3D substructure representations into a specialized 3D token vocabulary derived from the E3FP algorithm, the model effectively integrates sequence and structure representations in a tokenized format. The authors propose a multi-task pre-training scheme, including denoising and translation tasks, to enhance cross-modal comprehension and alignment of molecular sequences, spatial structures, and textual data.

A key strength of the paper is its innovative approach to integrating 3D structural data directly into a language model, addressing a crucial and often underexplored aspect of molecular modeling. The model removes dependency on separate structure encoders by incorporating 3D tokens at the atomic level, allowing for effective alignment between molecular sequences and 3D structures. The comprehensive evaluation across multiple tasks showcases the model's robustness and generalization capabilities, with notable improvements over current methods.

For the potential improvements, the detailed ablation studies would strengthen the model design and clarify the the individual contributions of each components. Evaluating the model on a broader range of datasets, including more diverse and complex ones like USPTO or QMUGS, would enhance the conclusiveness of the results.

Overall, I recommend the acceptance of this paper.

**Additional Comments On Reviewer Discussion:**

During the discussion period, all reviewers make the response and update the score.

---

### Decision · Program_Chairs · 2025-01-22

Accept (Poster)